# DIFFormer: Scalable (Graph) Transformers Induced by Energy Constrained Diffusion

**Qitian Wu[†], Chenxiao Yang[†], Wentao Zhao[†], Yixuan He[‡], David Wipf[§], Junchi Yan[†*]**
† Department of CSE & MoE Lab of Artificial Intelligence, Shanghai Jiao Tong University
‡ Department of Statistics, University of Oxford
§ Amazon Web Service
{echo740,chr26195,permanent,yanjunchi}@sjtu.edu.cn,
yixuan.he@stats.ox.ac.uk, davidwipf@gmail.com

## Abstract

Real-world data generation often involves complex inter-dependencies among instances, violating the IID-data hypothesis of standard learning paradigms and posing a challenge for uncovering the geometric structures for learning desired instance representations. To this end, we introduce an energy constrained diffusion model which encodes *a batch of instances* from a dataset into evolutionary states that progressively incorporate other instances' information by their interactions. The diffusion process is constrained by descent criteria w.r.t. a principled energy function that characterizes the global consistency of instance representations over latent structures. We provide rigorous theory that implies closed-form optimal estimates for the pairwise diffusion strength among arbitrary instance pairs, which gives rise to a new class of neural encoders, dubbed as DIFFormer (diffusion-based Transformers), with two instantiations: a simple version with linear complexity for prohibitive instance numbers, and an advanced version for learning complex structures. Experiments highlight the wide applicability of our model as a general-purpose encoder backbone with superior performance in various tasks, such as node classification on large graphs, semi-supervised image/text classification, and spatial-temporal dynamics prediction. The codes are available at https://github.com/qitianwu/DIFFormer.

## 1 Introduction

Real-world data are generated from a convoluted interactive process whose underlying physical principles are often unknown. Such a nature violates the common hypothesis of standard representation learning paradigms assuming that data are IID sampled. The challenge, however, is that due to the absence of prior knowledge about ground-truth data generation, it can be practically prohibitive to build feasible methodology for uncovering data dependencies, despite the acknowledged significance. To address this issue, prior works, e.g., Wang et al. (2019); Franceschi et al. (2019); Jiang et al. (2019); Zhang et al. (2019), consider encoding the potential interactions between instance pairs, but this requires sufficient degrees of freedom that significantly increases learning difficulty from limited labels (Fatemi et al., 2021) and hinders the scalability to large systems (Wu et al., 2022b).

Turning to a simpler problem setting where putative instance relations are instantiated as an observed graph, remarkable progress has been made in designing expressive architectures such as graph neural networks (GNNs) (Scarselli et al., 2008; Kipf & Welling, 2017; Velickovic et al., 2018; Wu et al., 2019; Chen et al., 2020a; Yang et al., 2021) for harnessing inter-connections between instances as a geometric prior (Bronstein et al., 2017). However, the observed relations can be incomplete/noisy, due to error-prone data collection, or generated by an artificial construction independent from downstream targets. The potential inconsistency between observation and the underlying data geometry would presumably elicit systematic bias between structured representation of graph-based learning and true

---

*Corresponding author: Junchi Yan who is also affiliated with Shanghai AI Lab. The work was in part supported by National Key Research and Development Program of China (2020AAA0107600), National Natural Science Foundation of China (62222607), STCSM (22511105100).

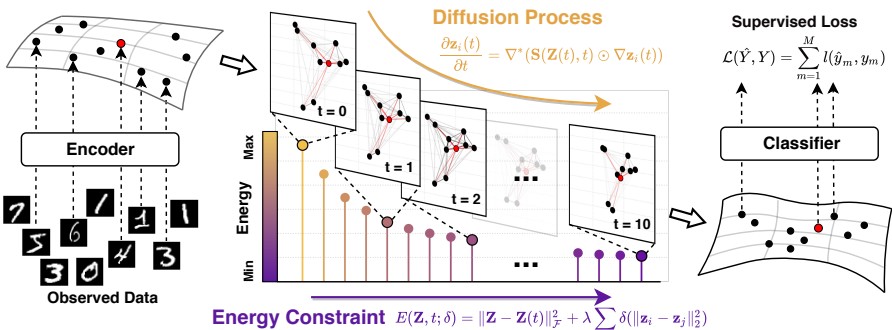

Figure 1: An illustration of the general idea behind DIFFORMER which takes a whole dataset (or a batch) of instances as input and encodes them into hidden states through a diffusion process aimed at minimizing a regularized energy. This design allows feature propagation among arbitrary instance pairs at each layer with optimal inter-connecting structures for informed prediction on each instance.

data dependencies. While a plausible remedy is to learn more useful structures from the data, this unfortunately brings the previously-mentioned obstacles to the fore.

To resolve the dilemma, we propose a novel general-purpose encoder framework that uncovers data dependencies from observations (a dataset of partially labeled instances), proceeding via two-fold inspiration from physics as illustrated in Fig. 1. Our model is defined through feed-forward continuous dynamics (i.e., a PDE) involving all the instances of a dataset as locations on Riemannian manifolds with *latent* structures, upon which the features of instances act as heat flowing over the underlying geometry (Hamzi & Owhadi, 2021). Such a diffusion model serves an important *inductive bias* for leveraging global information from other instances to obtain more informative representations. Its major advantage lies in the flexibility for the *diffusivity* function, i.e., a measure of the rate at which information spreads (Rosenberg & Steven, 1997): we allow for feature propagation between arbitrary instance pairs at each layer, and adaptively navigate this process by pairwise connectivity weights. Moreover, for guiding the instance representations towards some ideal constraints of internal consistency, we introduce a principled energy function that enforces layer-wise *regularization* on the evolutionary directions. The energy function provides another view (from a macroscopic standpoint) into the desired instance representations with low energy that are produced, i.e., soliciting a steady state that gives rise to informed predictions on unlabeled data.

As a justification for the tractability of above general methodology, our theory reveals the underlying equivalence between finite-difference iterations of the diffusion process and unfolding the minimization dynamics for an associated regularized energy. This result further suggests a closed-form optimal solution for the diffusivity function that updates instance representations by the ones of all the other instances towards giving a rigorous decrease of the global energy. Based on this, we also show that the energy constrained diffusion model can serve as a principled perspective for unifying popular models like MLP, GCN and GAT which can be viewed as special cases of our framework.

On top of the theory, we propose a new class of neural encoders, Diffusion-based Transformers (DIFFORMER), and its two practical instantiations: one is a simple version with $\mathcal{O}(N)$ complexity ($N$ for instance number) for computing all-pair interactions among instances; the other is a more expressive version that can learn complex latent structures. We empirically demonstrate the success of DIFFORMER on a diverse set of tasks. It outperforms SOTA approaches on semi-supervised node classification benchmarks and performs competitively on large-scale graphs. It also shows promising power for image/text classification with low label rates and predicting spatial-temporal dynamics.

## 2 RELATED WORK

**Graph-based Semi-supervised Learning.** Graph-based SSL (Kipf & Welling, 2017) aims to learn from partially labeled data, where instances are treated as nodes and their relations are given by a graph. The observed structure can be leveraged as regularization for learning representations (Belkin et al., 2006; Weston et al., 2012; Yang et al., 2016) or as an inductive bias of modern GNN architectures (Scarselli et al., 2008). However, there frequently exist situations where the observed structure is unavailable or unreliable (Franceschi et al., 2019; Jiang et al., 2019; Chen et al., 2020c; Fatemi et al., 2021; Lao et al., 2022), in which case the challenge remains how to uncover the underlying

relations. This paper explores a new Transformer-like encoder for discovering data geometry to promote learning through the inter-dependence among instances (either labeled or unlabeled).

**Neural Diffusion Models.** Several recent efforts explore diffusion-based learning where continuous dynamics serve as an inductive bias for representation learning (Hamzi & Owhadi, 2021). These works can be generally grouped into PDE-based learning, where the model itself is a continuous diffusion process described by a differential equation (e.g., Chamberlain et al. (2021a); Eliasof et al. (2021); Thorpe et al. (2022)), and PDE-inspired learning, where the diffusion perspective is adopted to develop new models (Atwood & Towsley, 2016; Klicpera et al., 2019) or characterize graph geometric properties (Yang et al., 2022). Our work leans on the later category and the key originality lies in two aspects. First, we introduce a novel diffusion model whose dynamics are implicitly defined by optimizing a regularized energy. Second, our theory establishes an equivalence between the numerical iterations of diffusion process and unfolding the optimization of the energy, based on which we develop a new class of neural encoders for uncovering latent structures among a large number of instances.

**Transformers.** Transformers serve as a model class of wide research interest showing competitive efficacy for modeling the dependency among tokens of inputs through all-pair attention mechanism. While the original architecture (Vaswani et al., 2017) is motivated by the inter-dependence among words of a sentence in NLP tasks, from different aspects, this work targets a Transformer-like architecture for modeling the inter-dependence among instances in a dataset. In the terminology of Transformers in NLPs, one can treat the whole dataset as an extremely long 'sequence' (with the length equaling to the dataset size), and each instance (with a label to predict) acts as the 'token' in such a sequence. In the context of graph machine learning, our goal can be framed as learning latent interaction graphs among nodes beyond the observed graph (if available), and this can be generally viewed as an embodiment of node-level prediction (Hu et al., 2020). For the latter case, critically though, it remains under-explored how to build a scalable and expressive Transformer for learning node-pair interactions given the prohibitively large number of node instances (Wu et al., 2022b).

## 3 ENERGY CONSTRAINED GEOMETRIC DIFFUSION TRANSFORMERS

Consider a set of partially labeled instances $\{\mathbf{x}_i\}_{i=1}^N$, whose labeled portion is $\{(\mathbf{x}_j, y_j)\}_{j=1}^M$ (often $M \ll N$). In some cases there exist relational structures that connect instances as a graph $\mathcal{G} = (\mathcal{V}, \mathcal{E})$, where the node set $\mathcal{V}$ contains all the instances and the edge set $\mathcal{E} = \{e_{ij}\}$ consists of observed relations. Without loss of generality, the main body of this section does *not* assume graph structures as input, but we will later discuss how to trivially incorporate them if they are available.

### 3.1 GEOMETRIC DIFFUSION MODEL

The starting point of our model is a diffusion process that treats a dataset of instances as a whole and produces instance representations through information flows characterized by an anisotropic diffusion process, which is inspired by an analogy with heat diffusion on a Riemannian manifold (Rosenberg & Steven, 1997). We use a vector-valued function $\mathbf{z}_i(t) : [0, \infty) \to \mathbb{R}^d$ to define an instance's state at time $t$ and location $i$. The anisotropic diffusion process describes the evolution of instance states (i.e., representations) via a PDE with boundary conditions (Freidlin & Wentzell, 1993; Medvedev, 2014):

$$\frac{\partial \mathbf{Z}(t)}{\partial t} = \nabla^* \left( \mathbf{S}(\mathbf{Z}(t), t) \odot \nabla \mathbf{Z}(t) \right), \quad \text{s. t. } \mathbf{Z}(0) = [\mathbf{x}_i]_{i=1}^N, \ t \geq 0, \tag{1}$$

where $\mathbf{Z}(t) = [\mathbf{z}_i(t)]_{i=1}^N \in \mathbb{R}^{N \times d}$, $\odot$ denotes the Hadamard product, and the function $\mathbf{S}(\mathbf{Z}(t), t) : \mathbb{R}^{N \times d} \times [0, \infty) \to [0, 1]^{N \times N}$ defines the *diffusivity* coefficient controlling the diffusion strength between any pair at time $t$. The diffusivity is specified to be dependent on instances' states. The gradient operator $\nabla$ measures the difference between source and target states, i.e., $(\nabla \mathbf{Z}(t))_{ij} = \mathbf{z}_j(t) - \mathbf{z}_i(t)$, and the divergence operator $\nabla^*$ sums up information flows at a point, i.e., $(\nabla^*)_i = \sum_{j=1}^N \mathbf{S}_{ij}(\mathbf{Z}(t), t) (\nabla \mathbf{Z}(t))_{ij}$. Note that both operators are defined over a discrete space consisting of $N$ locations. The physical implication of Eq. 1 is that the temporal change of heat at location $i$ equals to the heat flux that spatially enters into the point. Eq. 1 can be explicitly written as

$$\frac{\partial \mathbf{z}_i(t)}{\partial t} = \sum_{j=1}^N \mathbf{S}_{ij}(\mathbf{Z}(t), t)(\mathbf{z}_j(t) - \mathbf{z}_i(t)). \tag{2}$$

Such a diffusion process can serve as an inductive bias that guides the model to use other instances' information at every layer for learning informative instance representations. We can use numerical methods to solve the continuous dynamics in Eq. 2, e.g., the explicit Euler scheme involving finite differences with step size $\tau$, which after some re-arranging gives:

$$\mathbf{z}_i^{(k+1)} = \left(1 - \tau \sum_{j=1}^{N} \mathbf{S}_{ij}^{(k)}\right) \mathbf{z}_i^{(k)} + \tau \sum_{j=1}^{N} \mathbf{S}_{ij}^{(k)} \mathbf{z}_j^{(k)}. \tag{3}$$

The numerical iteration can stably converge for $\tau \in (0, 1)$. We can adopt the state after a finite number $K$ of propagation steps and use it for final predictions, i.e., $\hat{y}_i = \text{MLP}(\mathbf{z}_i^{(K)})$.

***Remark.*** The diffusivity coefficient in Eq. 1 is a measure of the rate at which heat can spread over the space (Rosenberg & Steven, 1997). Particularly in Eq. 2, $\mathbf{S}(\mathbf{Z}(t), t)$ determines how information flows over instances and the evolutionary direction of instance states. Much flexibility remains for its specification. For example, a basic choice is to fix $\mathbf{S}(\mathbf{Z}(t), t)$ as an identity matrix which constrains the feature propagation to self-loops and the model degrades to an MLP that treats all the instances independently. One could also specify $\mathbf{S}(\mathbf{Z}(t), t)$ as the observed graph structure if available in some scenarios. In such a case, however, the information flows are restricted by neighboring nodes in a graph. An ideal case could be to allow $\mathbf{S}(\mathbf{Z}(t), t)$ to have non-zero values for arbitrary $(i, j)$ and evolve with time, i.e., the instance states at each layer can efficiently and adaptively propagate to all the others.

## 3.2 Diffusion Constrained by a Layer-wise Energy

As mentioned previously, the crux is how to define a proper diffusivity function to induce a desired diffusion process that can maximize the information utility and accord with some inherent consistency. Since we have no prior knowledge for the explicit form or the inner structure of $\mathbf{S}^{(k)}$, we consider the diffusivity as a time-dependent latent variable and introduce an *energy function* that measures the presumed quality of instance states at a given step $k$:

$$E(\mathbf{Z}, k; \delta) = \|\mathbf{Z} - \mathbf{Z}^{(k)}\|_{\mathcal{F}}^2 + \lambda \sum_{i,j} \delta(\|\mathbf{z}_i - \mathbf{z}_j\|_2^2), \tag{4}$$

where $\delta : \mathbb{R}^+ \to \mathbb{R}$ is defined as a function that is *non-decreasing* and *concave* on a particular interval of our interest, and promotes robustness against large differences (Yang et al., 2021) among any pair of instances. Eq. 4 assigns each state in $\mathbb{R}^d$ with an energy scalar which can be leveraged to regularize the updated states (towards lower energy desired). The weight $\lambda$ trades two effects: 1) for each instance $i$, the states not far from the current one $\mathbf{z}_i^{(k)}$ have low energy; 2) for all instances, the smaller differences their states have, the lower energy is produced.

***Remark.*** Eq. 4 can essentially be seen as a robust version of the energy introduced by Zhou et al. (2004), inheriting the spirit of regularizing the global and local consistency of representations. "Robust" here particularly implies that the $\delta$ adds uncertainty to each pair of the instances and could *implicitly* filter the information of noisy links (potentially reflected by proximity in the latent space).

**Energy Constrained Diffusion.** The diffusion process describes the *microscopic* behavior of instance states through evolution, while the energy function provides a *macroscopic* view for quantifying the consistency. In general, we expect that the final states could yield a low energy, which suggests that the physical system arrives at a steady point wherein the yielded instance representations have absorbed enough global information under a certain guiding principle. Thereby, we unify two schools of thoughts into a new diffusive system where instance states would evolve towards producing lower energy, e.g., by finding a valid diffusivity function. Formally, we aim to find a series of $\mathbf{S}^{(k)}$'s whose dynamics and constraints are given by

$$\mathbf{z}_i^{(k+1)} = \left(1 - \tau \sum_{j=1}^{N} \mathbf{S}_{ij}^{(k)}\right) \mathbf{z}_i^{(k)} + \tau \sum_{j=1}^{N} \mathbf{S}_{ij}^{(k)} \mathbf{z}_j^{(k)}$$

$$\text{s. t. } \mathbf{z}_i^{(0)} = \mathbf{x}_i, \quad E(\mathbf{Z}^{(k+1)}, k; \delta) \leq E(\mathbf{Z}^{(k)}, k-1; \delta), \quad k \geq 1. \tag{5}$$

The formulation induces a new class of geometric flows on latent manifolds whose dynamics are *implicitly* defined by optimizing a time-varying energy function (see Fig. 1 for an illustration).

### 3.3 Tractability of Solving Diffusion Process with Energy Minimization

Unfortunately, Eq. 5 is hard to solve since we need to infer the value for a series of coupled $\mathbf{S}^{(k)}$'s that need to satisfy $K$ inequalities by the energy minimization constraint. The key result of this paper is the following theorem that reveals the underlying connection between the geometric diffusion model and iterative minimization of the energy, which further suggests an explicit closed-form solution for $\mathbf{S}^{(k)}$ based on the current states $\mathbf{Z}^{(k)}$ that yields a rigorous decrease of the energy.

**Theorem 1.** *For any regularized energy defined by Eq. 4 with a given $\lambda$, there exists $0 < \tau < 1$ such that the diffusion process of Eq. 3 with the diffusivity between pair $(i, j)$ at the $k$-th step given by*

$$\hat{\mathbf{S}}_{ij}^{(k)} = \frac{\omega_{ij}^{(k)}}{\sum_{l=1}^{N} \omega_{il}^{(k)}}, \quad \omega_{ij}^{(k)} = \left. \frac{\partial \delta(z^2)}{\partial z^2} \right|_{z^2 = \|\mathbf{z}_i^{(k)} - \mathbf{z}_j^{(k)}\|_2^2}, \tag{6}$$

*yields a descent step on the energy, i.e., $E(\mathbf{Z}^{(k+1)}, k; \delta) \leq E(\mathbf{Z}^{(k)}, k - 1; \delta)$ for any $k \geq 1$.*

Theorem 1 suggests the existence for the optimal diffusivity in the form of a function over the $l_2$ distance between states at the current step, i.e., $\|\mathbf{z}_i^{(k)} - \mathbf{z}_j^{(k)}\|_2$. The result enables us to unfold the implicit process and compute $\mathbf{S}^{(k)}$ in a feed-forward way from the initial states. We thus arrive at a new family of neural model architectures with layer-wise computation specified by:

$$\text{Diffusivity Inference:} \quad \hat{\mathbf{S}}_{ij}^{(k)} = \frac{f(\|\mathbf{z}_i^{(k)} - \mathbf{z}_j^{(k)}\|_2^2)}{\sum_{l=1}^{N} f(\|\mathbf{z}_i^{(k)} - \mathbf{z}_l^{(k)}\|_2^2)}, \quad 1 \leq i, j \leq N,$$

$$\text{State Updating:} \quad \mathbf{z}_i^{(k+1)} = \underbrace{\left(1 - \tau \sum_{j=1}^{N} \hat{\mathbf{S}}_{ij}^{(k)}\right) \mathbf{z}_i^{(k)}}_{\text{state conservation}} + \underbrace{\tau \sum_{j=1}^{N} \hat{\mathbf{S}}_{ij}^{(k)} \mathbf{z}_j^{(k)}}_{\text{state propagation}}, \quad 1 \leq i \leq N. \tag{7}$$

***Remark.*** The choice of function $f$ in above formulation is not arbitrary, but needs to be a non-negative and decreasing function of $z^2$, so that the associated $\delta$ in Eq. 4 is guaranteed to be non-decreasing and concave w.r.t. $z^2$. Critically though, there remains much room for us to properly design the specific $f$, so as to provide adequate capacity and scalability. Also, in our model presented by Eq. 7 we only have one hyper-parameter $\tau$ in practice, noting that the weight $\lambda$ in the regularized energy is implicitly determined through $\tau$ by Theorem 1, which reduces the cost of hyper-parameter searching.

## 4 Instantiations of DIFFormer

### 4.1 Model Instantiations

We next go into model instantiations based on the above theory, with two specified $f$'s as practical versions of our model. Due to space limits, we describe the key ideas concerning the model design in this subsection, and defer the details of model architectures to the self-contained Appendix E. First, because $\|\mathbf{z}_i - \mathbf{z}_j\|_2^2 = \|\mathbf{z}_i\|_2^2 + \|\mathbf{z}_j\|_2^2 - 2\mathbf{z}_i^\top \mathbf{z}_j$, we can convert $f(\|\mathbf{z}_i - \mathbf{z}_j\|_2^2)$ into the form $g(\mathbf{z}_i^\top \mathbf{z}_j)$ using a change of variables on the condition that $\|\mathbf{z}_i\|_2$ remains constant. And we add layer normalization to each layer to loosely enforce such a property in practice.

**Simple Diffusivity Model.** A straightforward design is to adopt the linear function $g(x) = 1 + x$:

$$\omega_{ij}^{(k)} = f(\|\tilde{\mathbf{z}}_i^{(k)} - \tilde{\mathbf{z}}_j^{(k)}\|_2^2) = 1 + \left(\frac{\mathbf{z}_i^{(k)}}{\|\mathbf{z}_i^{(k)}\|_2}\right)^\top \left(\frac{\mathbf{z}_j^{(k)}}{\|\mathbf{z}_j^{(k)}\|_2}\right), \tag{8}$$

Assuming $\tilde{\mathbf{z}}_i^{(k)} = \frac{\mathbf{z}_i^{(k)}}{\|\mathbf{z}_i^{(k)}\|_2}, \tilde{\mathbf{z}}_j^{(k)} = \frac{\mathbf{z}_j^{(k)}}{\|\mathbf{z}_j^{(k)}\|_2}$ and $z = \|\tilde{\mathbf{z}}_i^{(k)} - \tilde{\mathbf{z}}_j^{(k)}\|_2$, Eq. 8 can be written as $f(z^2) = 2 - \frac{1}{2}z^2$, which yields a non-negative result and is decreasing on the interval $[0, 2]$ in which $z^2$ lies. One scalability concern for the model Eq. 7 arises because of the need to compute pairwise diffusivity and propagation for each individual, inducing $\mathcal{O}(N^2)$ complexity. Remarkably, the simple diffusivity model allows a significant acceleration by noting that the state propagation can be re-arranged via

$$\sum_{j=1}^{N} \mathbf{S}_{ij}^{(k)} \mathbf{z}_j^{(k)} = \sum_{j=1}^{N} \frac{1 + (\tilde{\mathbf{z}}_i^{(k)})^\top \tilde{\mathbf{z}}_j^{(k)}}{\sum_{l=1}^{N} \left(1 + (\tilde{\mathbf{z}}_i^{(k)})^\top \tilde{\mathbf{z}}_l^{(k)}\right)} \mathbf{z}_j^{(k)} = \frac{\sum_{j=1}^{N} \mathbf{z}_j^{(k)} + \left(\sum_{j=1}^{N} \tilde{\mathbf{z}}_j^{(k)} \cdot (\mathbf{z}_j^{(k)})^\top\right) \cdot \tilde{\mathbf{z}}_i^{(k)}}{N + (\tilde{\mathbf{z}}_i^{(k)})^\top \sum_{l=1}^{N} \tilde{\mathbf{z}}_l^{(k)}}. \tag{9}$$

Table 1: A unified view for MLP, GCN and GAT from our energy-driven geometric diffusion framework regarding energy function forms, diffusivity specifications and algorithmic complexity.

| Models | Energy Function $E(\mathbf{Z}, k; \delta)$ | Diffusivity $\mathbf{S}^{(k)}$ | Complexity |
|---|---|---|---|
| MLP | $\|\mathbf{Z} - \mathbf{Z}^{(k)}\|_2^2$ | $\mathbf{S}_{ij}^{(k)} = \begin{cases} 1, & \text{if } i = j \\ 0, & otherwise \end{cases}$ | $\mathcal{O}(NKd^2)$ |
| GCN | $\sum_{(i,j)\in\mathcal{E}} \|\mathbf{z}_i - \mathbf{z}_j\|_2^2$ | $\mathbf{S}_{ij}^{(k)} = \begin{cases} \dfrac{1}{\sqrt{d_i d_j}}, & \text{if } (i,j)\in\mathcal{E} \\ 0, & otherwise \end{cases}$ | $\mathcal{O}(|\mathcal{E}|Kd^2)$ |
| GAT | $\sum_{(i,j)\in\mathcal{E}} \delta(\|\mathbf{z}_i - \mathbf{z}_j\|_2^2)$ | $\mathbf{S}_{ij}^{(k)} = \begin{cases} \dfrac{f(\|\mathbf{z}_i^{(k)} - \mathbf{z}_j^{(k)}\|_2^2)}{\sum_{l:(i,l)\in\mathcal{E}} f(\|\mathbf{z}_i^{(k)} - \mathbf{z}_l^{(k)}\|_2^2)}, & \text{if } (i,j)\in\mathcal{E} \\ 0, & otherwise \end{cases}$ | $\mathcal{O}(|\mathcal{E}|Kd^2)$ |
| DIFFORMER | $\|\mathbf{Z} - \mathbf{Z}^{(k)}\|_2^2 + \lambda\sum_{i,j} \delta(\|\mathbf{z}_i - \mathbf{z}_j\|_2^2)$ | $\mathbf{S}_{ij}^{(k)} = \dfrac{f(\|\mathbf{z}_i^{(k)} - \mathbf{z}_j^{(k)}\|_2^2)}{\sum_{l=1}^{N} f(\|\mathbf{z}_i^{(k)} - \mathbf{z}_l^{(k)}\|_2^2)}, \quad 1 \le i, j \le N$ | DIFFORMER-s: $\mathcal{O}(NKd^2)$ 
 DIFFORMER-a: $\mathcal{O}(N^2Kd^2)$ |

The two summation terms above can be computed once and shared to every instance $i$, reducing the complexity in each iteration to $\mathcal{O}(N)$. We refer to this implementation as DIFFORMER-s.

**Advanced Diffusivity Model.** The simple model facilitates efficiency/scalability, yet may sacrifice the capacity for complex latent geometry. We thus propose an advanced version with $g(x) = \frac{1}{1+\exp(-x)}$:

$$\omega_{ij}^{(k)} = f(\|\tilde{\mathbf{z}}_i^{(k)} - \tilde{\mathbf{z}}_j^{(k)}\|_2^2) = \frac{1}{1 + \exp\left(-(\mathbf{z}_i^{(k)})^\top(\mathbf{z}_j^{(k)})\right)}, \tag{10}$$

which corresponds with $f(z^2) = \frac{1}{1+e^{z^2/2-1}}$ guaranteeing monotonic decrease and non-negativity. We dub this version as DIFFORMER-a. Appendix D further compares the two models (i.e., different $f$'s and $\delta$'s) through synthetic results. Real-world empirical comparisons are in Section 5.

## 4.2 MODEL EXTENSIONS AND FURTHER DISCUSSION

**Incorporating Layer-wise Transformations.** Eq. 7 does not use feature transformations for each layer. To further improve the representation capacity, we can add such transformations after the updating, i.e., $\mathbf{z}_i^{(k)} \leftarrow h^{(k)}(\mathbf{z}_i^{(k)})$ where $h^{(k)}$ can be a fully-connected layer (see Appendix E for details). In this way, each iteration of the diffusion yields a descent of a particular energy $E(\mathbf{Z}, k; \delta, h^{(k)}) = \|\mathbf{Z} - h^{(k)}(\mathbf{Z}^{(k)})\|_2^2 + \sum_{i,j} \delta(\|\mathbf{z}_i - \mathbf{z}_j\|_2^2)$ dependent on $k$. The trainable transformation $h^{(k)}$ can be optimized w.r.t. the supervised loss to map the instance representations into a proper latent space. Our experiments find that the layer-wise transformation is not necessary for small datasets, but contributes to positive effects for datasets with larger sizes. Furthermore, one can consider non-linear activations in the layer-wise transformation $h^{(k)}$ though we empirically found that using a linear model already performs well. We also note that Theorem 1 can be extended to hold even when incorporating such a non-linearity in each layer (see Appendix C for detailed discussions).

**Incorporating Input Graphs.** For the model presented so far, we do *not* assume an input graph for the model formulation. For situations with observed structures as available input, we have $\mathcal{G} = (\mathcal{V}, \mathcal{E})$ that can be leveraged as a geometric prior. We can thus modify the updating rule as:

$$\mathbf{z}_i^{(k+1)} = \left(1 - \frac{\tau}{2}\sum_{j=1}^{N}\left(\hat{\mathbf{S}}_{ij}^{(k)} + \tilde{\mathbf{A}}_{ij}\right)\right)\mathbf{z}_i^{(k)} + \frac{\tau}{2}\sum_{j=1}^{N}\left(\hat{\mathbf{S}}_{ij}^{(k)} + \tilde{\mathbf{A}}_{ij}\right)\mathbf{z}_j^{(k)}, \tag{11}$$

where $\tilde{\mathbf{A}}_{ij} = \frac{1}{\sqrt{d_i d_j}}$ if $(i,j) \in \mathcal{E}$ and 0 otherwise, and $d_i$ is instance $i$'s degree in $\mathcal{G}$. The diffusion iteration of Eq. 11 is essentially a descent step on a new energy additionally incorporating a graph-based penalty (Ioannidis et al., 2017), i.e., $\sum_{(i,j)\in\mathcal{E}} \|\mathbf{z}_i^{(k)} - \mathbf{z}_j^{(k)}\|_2^2$ (see Appendix D for details).

**Scaling to Large Datasets.** Another advantage of DIFFORMER over GNNs is the flexibility for mini-batch training. For datasets with prohibitive instance numbers that make it hard for full-batch training on a single GPU, the common practice for GNNs resorts to subgraph sampling Zeng et al. (2020) or graph clustering Chiang et al. (2019) to reduce the overhead. These strategies, however, require extra time and tricky designs to preserve the internal structures. In contrast, thanks to the less reliance on input graphs, for training DIFFORMER, we can naturally partition the dataset into random mini-batches and feed one mini-batch for one feed-forward and backward computation. The

Table 2: Mean and standard deviation of testing accuracy on node classification (with five different random initializations). All the models are split into groups with a comparison of non-linearity (whether the model requires activation for layer-wise transformations), PDE-solver (whether the model requires PDE-solver) and Input-G (whether the propagation purely relies on input graphs).

| Type | Model | Non-linearity | PDE-solver | Input-G | Cora | Citeseer | Pubmed |
|---|---|---|---|---|---|---|---|
| Basic models | MLP | R | - | - | $56.1 \pm 1.6$ | $56.7 \pm 1.7$ | $69.8 \pm 1.5$ |
| | LP | - | - | R | 68.2 | 42.8 | 65.8 |
| | ManiReg | R | - | R | $60.4 \pm 0.8$ | $67.2 \pm 1.6$ | $71.3 \pm 1.4$ |
| Standard GNNs | GCN | R | - | R | $81.5 \pm 1.3$ | $71.9 \pm 1.9$ | $77.8 \pm 2.9$ |
| | GAT | R | - | R | $83.0 \pm 0.7$ | $72.5 \pm 0.7$ | $79.0 \pm 0.3$ |
| | SGC | - | - | R | $81.0 \pm 0.0$ | $71.9 \pm 0.1$ | $78.9 \pm 0.0$ |
| | GCN-$k$NN | R | - | - | $72.2 \pm 1.8$ | $56.8 \pm 3.2$ | $74.5 \pm 3.2$ |
| | GAT-$k$NN | R | - | - | $73.8 \pm 1.7$ | $56.4 \pm 3.8$ | $75.4 \pm 1.3$ |
| | Dense GAT | R | - | - | $78.5 \pm 2.5$ | $66.4 \pm 1.5$ | $66.4 \pm 1.5$ |
| | LDS | R | - | - | $83.9 \pm 0.6$ | $74.8 \pm 0.3$ | out-of-memory |
| | GLCN | R | - | - | $83.1 \pm 0.5$ | $72.5 \pm 0.9$ | $78.4 \pm 1.5$ |
| Diffusion-based models | GRAND-l | - | R | R | $83.6 \pm 1.0$ | $73.4 \pm 0.5$ | $78.8 \pm 1.7$ |
| | GRAND | R | R | R | $83.3 \pm 1.3$ | $74.1 \pm 1.7$ | $78.1 \pm 2.1$ |
| | GRAND++ | R | R | R | $82.2 \pm 1.1$ | $73.3 \pm 0.9$ | $78.1 \pm 0.9$ |
| | GDC | R | - | R | $83.6 \pm 0.2$ | $73.4 \pm 0.3$ | $78.7 \pm 0.4$ |
| | GraphHeat | R | - | R | 83.7 | 72.5 | 80.5 |
| | DGC-Euler | - | - | R | $83.3 \pm 0.0$ | $73.3 \pm 0.1$ | $80.3 \pm 0.1$ |
| Graph Transformers | NodeFormer | - | - | - | $83.4 \pm 0.2$ | $73.0 \pm 0.3$ | $81.5 \pm 0.4$ |
| | DIFFORMER-s | - | - | - | $85.9 \pm 0.4$ | $73.5 \pm 0.3$ | $81.8 \pm 0.3$ |
| | DIFFORMER-a | - | - | - | $84.1 \pm 0.6$ | $75.7 \pm 0.3$ | $80.5 \pm 1.2$ |

flexibility for mini-batch training also brings up convenience for parallel acceleration and federated learning. In fact, such a computational advantage is shared by Transformer-like models for cases where instances are inter-dependent, as demonstrated by the NodeFormer (Wu et al., 2022b).

**Connection with Existing Models.** Our theory in fact gives rise to a general diffusion framework that unifies some existing models as special cases of ours. As a non-exhaustive summary and high-level comparison, Table 1 presents the relationships with MLP, GCN and GAT (see more elaboration in Appendix F). Specifically, the multi-layer perceptrons (MLP) can be seen as only considering the local consistency regularization in the energy function and only allowing non-zero diffusivity for self-loops. For graph convolution networks (GCN) (Kipf & Welling, 2017), it only regularizes the global consistency term constrained within 1-hop neighbors of the observed structure and simplifies $\delta$ as an identity function. The diffusivity of GCN induces information flows through observed edges. For graph attention networks (GAT) (Velickovic et al., 2018), the energy function is similar to GCN's except that the non-linearity $\delta$ remains (as a certain specific form), and the diffusivity is computed by attention over edges. In contrast with the above models, DIFFORMER is derived from the general diffusion model that enables non-zero diffusivity values between arbitrary instance pairs.

## 5 EXPERIMENTS

We apply DIFFORMER to various tasks for evaluation: 1) graph-based node classification where an input graph is given as observation; 2) image and text classification without input graphs; 3) spatial temporal dynamics prediction. In each case, we compare a different set of competing models closely associated with DIFFORMER and specifically designed for the particular task. Unless otherwise stated, for datasets where input graphs are available, we incorporate them for feature propagation as is defined by Eq. 11. Due to space limit, we defer details of datasets to Appendix G and the implementations to Appendix H. Also, we provide additional empirical results in Appendix I.

### 5.1 SEMI-SUPERVISED NODE CLASSIFICATION BENCHMARKS

We test DIFFORMER on three citation networks Cora, Citeseer and Pubmed. Table 2 reports the testing accuracy. We compare with several sets of baselines linked with our model from different aspects. 1) Basic models: *MLP* and two classical graph-based SSL models Label Propagation (*LP*) (Zhu et al., 2003) and *ManiReg* (Belkin et al., 2006). 2) GNN models: *SGC* (Wu et al., 2019), *GCN* (Kipf & Welling, 2017), *GAT* (Velickovic et al., 2018), their variants *GCN-kNN*, *GAT-kNN* (operating on $k$NN graphs constructed from input features) and *Dense GAT* (with a densely connected graph replacing the input one), and two strong structure learning models LDS (Franceschi et al., 2019) and GLCN (Jiang et al., 2019). 3) PDE graph models: the SOTA models *GRAND* (Chamberlain et al.,

Table 3: Testing ROC-AUC for `Proteins` and Accuracy for `Pokec` on large-scale node classification datasets. ∗ denotes using mini-batch training.

| Models | Proteins | Pokec |
|--------|----------|-------|
| MLP | $72.41 \pm 0.10$ | $60.15 \pm 0.03$ |
| LP | 74.73 | 52.73 |
| SGC | $49.03 \pm 0.93$ | $52.03 \pm 0.84$ |
| GCN | $74.22 \pm 0.49^*$ | $62.31 \pm 1.13^*$ |
| GAT | $75.11 \pm 1.45^*$ | $65.57 \pm 0.34^*$ |
| NodeFormer | $77.45 \pm 1.15^*$ | $68.32 \pm 0.45^*$ |
| DIFFORMER-s | $79.49 \pm 0.44^*$ | $69.24 \pm 0.76^*$ |

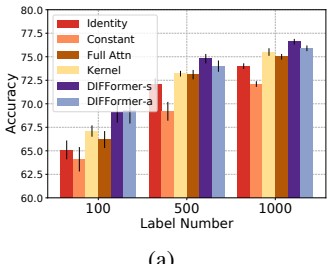

(a)

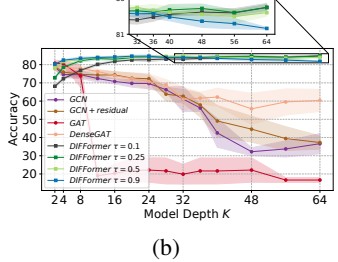

(b)

Figure 2: (a) Ablation studies w.r.t. different diffusivity function forms on `CIFAR`. (b) Impact of $K$ and $\tau$ on `Cora`.

Table 4: Testing accuracy for image (`CIFAR` and `STL`) and text (`20News`) classification.

| Dataset | | MLP | LP | ManiReg | GCN-$k$NN | GAT-$k$NN | DenseGAT | GLCN | DIFFORMER-s | DIFFORMER-a |
|---------|---|-----|----|---------|-----------|-----------|----------|------|-------------|-------------|
| **CIFAR** | 100 labels | $65.9 \pm 1.3$ | 66.2 | $67.0 \pm 1.9$ | $66.7 \pm 1.5$ | $66.0 \pm 2.1$ | out-of-memory | $66.6 \pm 1.4$ | $69.1 \pm 1.1$ | $69.3 \pm 1.4$ |
| | 500 labels | $73.2 \pm 0.4$ | 70.6 | $72.6 \pm 1.2$ | $72.9 \pm 0.4$ | $72.4 \pm 0.5$ | out-of-memory | $72.8 \pm 0.5$ | $74.8 \pm 0.5$ | $74.0 \pm 0.6$ |
| | 1000 labels | $75.4 \pm 0.6$ | 71.9 | $74.3 \pm 0.4$ | $74.7 \pm 0.5$ | $74.1 \pm 0.5$ | out-of-memory | $74.7 \pm 0.3$ | $76.6 \pm 0.3$ | $75.9 \pm 0.3$ |
| **STL** | 100 labels | $66.2 \pm 1.4$ | 65.2 | $66.5 \pm 1.9$ | $66.9 \pm 0.5$ | $66.5 \pm 0.8$ | out-of-memory | $66.4 \pm 0.8$ | $67.8 \pm 1.1$ | $66.8 \pm 1.1$ |
| | 500 labels | $73.0 \pm 0.8$ | 71.8 | $72.5 \pm 0.5$ | $72.1 \pm 0.8$ | $72.0 \pm 0.8$ | out-of-memory | $72.4 \pm 1.3$ | $73.7 \pm 0.6$ | $72.9 \pm 0.7$ |
| | 1000 labels | $75.0 \pm 0.8$ | 72.7 | $74.2 \pm 0.5$ | $73.7 \pm 0.4$ | $73.9 \pm 0.6$ | out-of-memory | $74.3 \pm 0.7$ | $76.4 \pm 0.5$ | $75.3 \pm 0.6$ |
| **20News** | 1000 labels | $54.1 \pm 0.9$ | 55.9 | $56.3 \pm 1.2$ | $56.1 \pm 0.6$ | $55.2 \pm 0.8$ | $54.6 \pm 0.2$ | $56.2 \pm 0.8$ | $57.7 \pm 0.3$ | $57.9 \pm 0.7$ |
| | 2000 labels | $57.8 \pm 0.9$ | 57.6 | $60.0 \pm 0.8$ | $60.6 \pm 1.3$ | $59.1 \pm 2.2$ | $59.3 \pm 1.4$ | $60.2 \pm 0.7$ | $61.2 \pm 0.6$ | $61.3 \pm 1.0$ |
| | 4000 labels | $62.4 \pm 0.6$ | 59.5 | $63.6 \pm 0.7$ | $64.3 \pm 1.0$ | $62.9 \pm 0.7$ | $62.4 \pm 1.0$ | $64.1 \pm 0.8$ | $65.9 \pm 0.8$ | $64.8 \pm 1.0$ |

2021a) (with its linear variant GRAND-l) and *GRAND++* (Thorpe et al., 2022). 4) Diffusion-inspired GNN models: *GDC* (Klicpera et al., 2019), *GraphHeat* (Xu et al., 2020) and a recent work *DGC-Euler* (Wang et al., 2021). Table 2 shows that DIFFORMER achieves the best results on three datasets with significant improvements. Also, we notice that the simple diffusivity model DIFFORMER-s significantly exceeds the counterparts without non-linearity (SGC, GRAND-l and DGC-Euler) and even comes to the first on `Cora` and `Pubmed`. These results suggest that DIFFORMER can serve as a very competitive encoder backbone for node-level prediction that learns inter-instance interactions for generating informative representations and boosting downstream performance.

## 5.2 LARGE-SCALE NODE CLASSIFICATION GRAPHS

We also consider two large-scale graph datasets `ogbn-Proteins`, a multi-task protein-protein interaction network, and `Pokec`, a social network. Table 3 presents the results. Due to the dataset size (0.13M/1.63M nodes for two graphs) and scalability issues that many of the competitors in Table 2 as well as DIFFORMER-a would potentially experience, we only compare DIFFORMER-s with standard GNNs. In particular, we found GCN/GAT/DIFFORMER-s are still hard for full-graph training on a single V100 GPU with 16GM memory. We thus consider mini-batch training with batch size 10K/100K for `Proteins`/`Pokec`. We found that DIFFORMER outperforms common GNNs by a large margin, which suggests its desired efficacy on large datasets. As mentioned previously, we prioritize the efficacy of DIFFORMER as a general encoder backbone for solving node-level prediction tasks on large graphs. While there are quite a few practical tricks shown to be effective for training GNNs for this purpose, e.g., hop-wise attention (Sun et al., 2022) or various label re-use strategies, these efforts are largely orthogonal to our contribution here and can be applied to most any model to further boost performance. For further investigation, we supplement more results using different mini-batch sizes for training and study its impact on testing performance in Appendix I.4. Furthermore, we compare the training time and memory costs in Appendix I.5 where we found that DIFFORMER-s is about 6 times faster than GAT and 39 times faster than DenseGAT on `Pokec`, which suggests superior scalability and efficiency of DIFFORMER-s on large graphs.

## 5.3 IMAGE AND TEXT CLASSIFICATION WITH LOW LABEL RATES

We next conduct experiments on `CIFAR-10`, `STL-10` and `20News-Group` datasets to test DIF-FORMER for standard classification tasks with limited label rates. For `20News` provided by Pedregosa et al. (2011), we take 10 topics and use words with TF-IDF more than 5 as features. For `CIFAR` and `STL`, two public image datasets, we first use the self-supervised approach SimCLR (Chen et al., 2020b) (that does not use labels for training) to train a ResNet-18 for extracting the feature maps as input features of instances. These datasets contain no graph structure, so we use $k$NN to construct a graph over input features for GNN competitors and do *not* use input graphs for DIFFORMER.

Table 5: Mean and standard deviation of MSE on spatial-temporal prediction datasets.

| Dataset | MLP | GCN | GAT | Dense GAT | GAT-$k$NN | GCN-$k$NN | DIFFORMER-s | DIFFORMER-a | DIFFORMER-s w/o g | DIFFORMER-a w/o g |
|---|---|---|---|---|---|---|---|---|---|---|
| **Chickenpox** | 0.924 (±0.001) | **0.923** (±**0.001**) | 0.924 (±0.002) | 0.935 (±0.005) | 0.926 (±0.004) | 0.936 (±0.004) | 0.926 (±0.002) | 0.926 (±0.008) | **0.920** (±**0.001**) | **0.920** (±**0.002**) |
| **Covid** | 0.956 (±0.198) | 1.080 (±0.162) | 1.052 (±0.336) | 1.524 (±0.319) | 0.861 (±0.123) | 1.475 (±0.560) | **0.792** (±**0.086**) | **0.792** (±**0.076**) | **0.791** (±**0.090**) | 0.935 (±0.143) |
| **WikiMath** | 1.073 (±0.042) | 1.292 (±0.125) | 1.339 (±0.073) | 0.826 (±0.070) | 0.882 (±0.015) | 1.023 (±0.058) | 0.922 (±0.015) | **0.738** (±**0.031**) | 0.993 (±0.042) | **0.720** (±**0.036**) |

Table 4 reports the testing accuracy of DIFFORMER and competitors including MLP, ManiReg, GCN-$k$NN, GAT-$k$NN, DenseGAT and GLCN. Two DIFFORMER models perform much better than MLP in nearly all cases, suggesting the effectiveness of learning the inter-dependence over instances. Besides, DIFFORMER yields large improvements over GCN and GAT which are in some sense limited by the handcrafted graph that leads to sub-optimal propagation. Moreover, DIFFORMER significantly outperforms GLCN, a strong baseline that learns new (static) graph structures, which demonstrates the superiority of our evolving diffusivity that can adapt to different layers.

### 5.4 SPATIAL-TEMPORAL DYNAMICS PREDICTION

We consider three spatial-temporal datasets with details in Appendix G. Each dataset consists of a series of graph snapshots where nodes are treated as instances and each of them has a integer label (e.g., reported cases for `Chickenpox` or `Covid`). The task is to predict the labels of one snapshot based on the previous ones. Table 5 compares testing MSE of four DIFFORMER variants (here DIFFORMER-s w/o g denotes the model DIFFORMER-s without using input graphs) against baselines. We can see that two DIFFORMER variants without input graphs even outperform the counterparts using input structures in four out of six cases. This implies that our diffusivity estimation module could learn useful structures for informed prediction, and the input structure might not always contribute to positive effect. In fact, for temporal dynamics, the underlying relations that truly influence the trajectory evolution can be much complex and the observed relations could be unreliable with missing or noisy links, in which case GNN models relying on input graphs may perform undesirably. Compared to the competitors, our models rank the first with significant improvements.

### 5.5 FURTHER RESULTS AND DISCUSSIONS

**How do different diffusivity functions perform?** Figure 2(a) compares DIFFORMER with four variants using other diffusivity functions that have no essential connection with energy minimization: 1) *Identity* sets $\mathbf{S}^{(k)}$ as a fixed identity matrix; 2) *Constant* fixes $\mathbf{S}^{(k)}$ as all-one constant matrix; 3) *Full Attn* parameterizes $\mathbf{S}^{(k)}$ by attention networks (Vaswani et al., 2017); 4) *Kernel* adopts Gaussian kernel for computing $\mathbf{S}^{(k)}$. More results on other datasets are in Appendix I.1 and they consistently show that our adopted diffusivity forms produce superior performance, which verifies the effectiveness of our diffusivity designs derived from minimization of a principled energy.

**How do model depth and step size impact the performance?** We discuss the influence of model depth $K$ and step size $\tau$ on `Cora` in Fig. 2(b). More results on `Citeseer` and `Pubmed` are generally consistent with Fig. 2(b) and deferred to Appendix I.2. The curves indicate that the performance of GCN and GAT models exhibit significant degradation with deeper layers, while DIFFORMER maintains its superiority and performs stably with large $K$. Furthermore, when $K$ is not large enough (less than 32), there is a clear performance improvement of DIFFORMER as $K$ increases, and larger $\tau$ contributes to a steeper increase. When $K$ continues increasing (more than 32), the model performance still goes up with small $\tau$ (0.1 and 0.25) yet exhibits a slight drop with large $\tau$ (0.5 and 0.9). The reason could be that larger (smaller) $\tau$ contributes to more (less) concentration on global information from other instances in each iteration, which brings up more (less) benefits with increasing propagation layers yet could lead to instability when the step size is too large.

## 6 CONCLUSIONS

This paper proposes an energy-driven geometric diffusion model with latent diffusivity function for data representations. The model encodes all the instances as a whole into evolving states aimed at minimizing a principled energy as implicit regularization. We further design two practical implementations with enough scalability and capacity for learning complex interactions over the underlying data geometry. Extensive experiments demonstrate the effectiveness and superiority of the model.

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

## A   FURTHER RELATED WORKS AND CONNECTION WITH OURS

We discuss more related works that associate with ours from different aspects to properly position this paper with different areas. Based on this, we further shed more lights on the technical contributions of our work and its potential impact in different communities.

### A.1   NEURAL DIFFUSION MODELS

**PDE-based Learning.** The diffusion-based learning has gained increasing research interests, as the continuous dynamics can serve as an inductive bias incorporated with prior knowledge of the tasks at hand. One category directly solves a continuous process of differential equations (Lagaris et al., 1998; Chen et al., 2018), e.g., Chamberlain et al. (2021a) and its follow-ups (Chamberlain et al., 2021b; Thorpe et al., 2022) reveal the analogy between the discretization of diffusion process and GNNs' feedforward rules, and devise new (continuous) models on graphs whose training requires PDE-solving tools. A concurrent work (Wang et al., 2023) explores how to derive neural networks from gradient flows and proposes Allen-Cahn Message Passing that combines attractive and repulsive effects.

**PDE-inspired Learning.** Another category is PDE-inspired learning using the diffusion perspective as principled guidelines on top of which new (discrete) neural network-based approaches are designed for node classification (Atwood & Towsley, 2016; Klicpera et al., 2019; Xu et al., 2020), graph comparison (Tsitsulin et al., 2018), and geometric knowledge distillation (Yang et al., 2022). Our work leans on PDE-inspired learning and introduce a new diffusion model that is implicitly defined as minimizing a regularized energy. Our theory also reveals the underlying equivalence between the numerical iterations of the diffusion process and unfolding the minimization dynamics of a corresponding energy. The new results can serve to extend the class of diffusion process and illuminate its fundamental connection with energy optimization systems. More importantly, as we will in the maintext, such a new perspective brings up a unifying view for existing models like MLP and GNNs.

## A.2 THEORETICAL INTERPRETATIONS OF GNNS

**Optimization-induced Models.** As GNNs have proven to be powerful encoders for modeling data with geometry, there arise quite a few studies that aim at understanding the efficacy of GNNs with respect to node representations yielded by the propagation. For instance, Yang et al. (2021) founds that the layer-wise updates of common GNNs can be derived from a classical iterative algorithm solving an optimization problem defined over an energy. Furthermore, Ma et al. (2021) points out the relationship between GNNs' message passing rules and the graph denoising problem. These works interpret GNNs as optimization-induced models. In our model, the diffusion process is implicitly defined by an optimization problem and our theory further reveals the fundamental equivalence between diffusion iterations and energy optimization dynamics, which can bridge two schools of thinking and facilitate the understandings from both perspectives.

**Generalization of GNNs.** Another line of works turn attention to GNNs' generalization capability. As increasing evidence shows that GNNs are sensitive to distribution shifts (Wu et al., 2022a), it becomes urgent to investigate and figure out how to improve GNNs' generalization. A concurrent work (Yang et al., 2023) identifies an intriguing phenomenon that GNNs without propagation (i.e., an MLP architecture) during training can still yield competitive or even close performance at inference time to standard GNNs (using feature propagation at both training and inference stage). This suggests that the propagation design contributes to better generalization instead of the model capacity, and the former plays as a critical factor that makes GNNs a more powerful encoder than MLP. This result can further interprets the success of DIFFORMER: the new propagation of DIFFORMER that aggregates other nodes' embeddings can enhance the generalization at inference time on testing data.

## A.3 TRANSFORMERS

**General Transformers.** The key design of Transformers lies in the (all-pair) attention mechanism that captures the interactions among tokens of an input sequence. The attention design is originally motivated by the inter-dependence among words of a sentence in NLP tasks (Vaswani et al., 2017). Furthermore, a surge of follow-up works extend the attention mechanisms for capturing the inter-dependence among patches of an image (Dosovitskiy et al., 2021), atoms of a molecule (Dwivedi & Bresson, 2020), dimensions of multivariate time series (Zhang & Yan, 2023), etc. From different aspects, our work explores a Transformer model that targets learning inter-dependence among instances in a dataset.

**Efficient Transformers.** When the length of inputs goes large, the quadratic complexity of all-pair attention becomes the computational bottleneck of Transformers. To address the long sequences in NLP tasks, quite a few recent works propose various strategies via, e.g., Softmax kernel (Choromanski et al., 2021), local-global attention (Zaheer et al., 2020), and low-rank approximation (Katharopoulos et al., 2020). In our problem setting, if one treats the whole dataset as an input 'sequence' for the Transformer model, then the length of such a sequence could go to even million-level, which poses demanding scalability challenges. Our model DIFFORMER-s achieves strictly linear complexity w.r.t. $N$ through introducing a new attention function that keeps a simple form yet accommodates the influence among arbitrary node pairs.

**Transformers on Graphs.** There is also increasing research interest on building powerful Transformers for graph-structured data, due to the inherent good ability of Transformer models for capturing long-range dependence and potential node-pair interactions that are unobserved in input structures.

Recent works, e.g., Dwivedi & Bresson (2020); Zhang et al. (2020); Ying et al. (2021) explore various effective architectures for graph-level prediction (GP) tasks, e.g., molecular property prediction, whereby each graph itself generally has a label to predict and a dataset contains many graph instances. Our target problem, as mentioned in Section 2, can be seen an embodiment of node-level prediction (NP) studied in graph learning community. These two problems are typically tackled separately in the literature (Hu et al., 2020) with disparate technical considerations. This is because input instances are inter-dependent in NP (due to the instance interactions involved in the data-generating process), while in GP tasks the instances can be treated as IID samples. For node-level prediction, a recent work (Wu et al., 2022b) proposes kernelized Gumbel-Softmax-based message passing to achieve all-pair feature propagation with linear complexity. Concurrently at the same conference, (Chen et al., 2023) proposes to transform neighborhood features into tokens fed into Transformers to enable efficient training on large graphs

### A.4 GRAPH NEURAL NETWORKS

**Scalable Graph Neural Networks.** Due to the large graph sizes that could be frequently encountered in graph-based predictive tasks (particularly NP), many recent works focus on designing scalable GNNs. From different technical aspects, these works can be generally grouped into graph sampling-based partition (Zeng et al., 2020; Chiang et al., 2019), approximation-based propagation (Bojchevski et al., 2020) and simplified architectures (Wu et al., 2019; Zhang et al., 2022). In contrast with them, DIFFORMER can scale to medium-sized graphs with the linear complexity, and to large-scale graphs with mini-batch training that is simple, stable and also flexible for balancing efficiency and precision with proper mini-batch sizes. The later is allowed by the all-pair message passing schemes that do not rely on input structures (the input graphs play an auxiliary role in the model).

**Graph Structure Learning.** Learning latent graph structures is a well-established research problem in graph learning community. Motivated by the observation that input graphs can often be unreliable or unavailable, based on which the message passing of GNNs could yield undesired results, graph structure learning aims at learning adaptive structures that can boost GNNs towards better representations and downstream prediction (Franceschi et al., 2019; Chen et al., 2020c; Jiang et al., 2019; Fatemi et al., 2021; Lao et al., 2022). Since graph structure learning requires estimation for $N^2$ potential edges that connect all node pairs in a dataset, the time and space complexity of most existing models are at least quadratic w.r.t. node numbers, which is prohibitive for large graphs. The recent work (Wu et al., 2022b) proposes a Transformer model with linear complexity that can scale graph structure learning to graphs with millions of nodes. The proposed model DIFFORMER-s can be another powerful approach for learning latent structures in large-scale systems, without sacrificing the accommodation of all-pair interactions at each layer.

## B PROOF FOR THEOREM 1

First of all, we can convert the minimization of Eq. 4 into a minimization of its variational upper bound, shown in the following proposition.

**Proposition 1.** *The energy function $E(\mathbf{Z}, k; \delta)$ is upper bounded by*

$$\tilde{E}(\mathbf{Z}, k; \{\omega_{ij}\}, \tilde{\delta}) = \|\mathbf{Z} - \mathbf{Z}^{(k)}\|_{\mathcal{F}}^2 + \lambda \left[ \sum_{i,j} \omega_{ij} \|\mathbf{z}_i - \mathbf{z}_j\|_2^2 - \tilde{\delta}(\omega_{ij}) \right], \quad (12)$$

*where $\tilde{\delta}$ is the concave conjugate of $\delta$, and the equality holds if and only if the variational parameters satisfy*

$$\omega_{ij} = \frac{\partial \delta(z^2)}{\partial z^2} \bigg|_{z=\|\mathbf{z}_i - \mathbf{z}_j\|_2}. \quad (13)$$

*Proof.* The proof of the proposition follows the principles of convex analysis and Fenchel duality (Rockafellar, 1970). For any concave and non-decreasing function $\rho : \mathbb{R}^+ \to \mathbb{R}$, one can express it as the variational decomposition

$$\rho(z^2) = \min_{\omega \geq 0}[\omega z^2 - \tilde{\rho}(\omega)] \geq \omega z^2 - \tilde{\rho}(\omega), \quad (14)$$

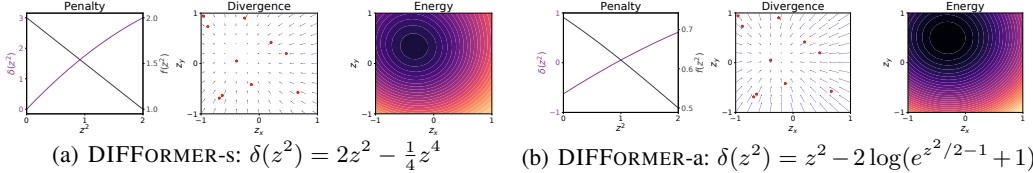

(a) DIFFORMER-s: $\delta(z^2) = 2z^2 - \frac{1}{4}z^4$     (b) DIFFORMER-a: $\delta(z^2) = z^2 - 2\log(e^{z^2/2-1}+1)$

Figure 3: Plot of penalty curves $\delta(z^2)$ and $f(z^2) = \frac{\partial \delta(z^2)}{\partial z^2}$, divergence field (produced by 10 randomly generated instances marked as red stars) and cross-section energy field of an individual.

where $\omega$ is a variational parameter and $\tilde{\rho}$ is the concave conjugate of $\rho$. Eq. 14 essentially defines $\rho(z^2)$ as the minimal envelope of a series of quadratic bounds $\omega z^2 - \tilde{\rho}(\omega)$ defined by a different values of $\omega \geq 0$ and the upper bound is given for a fixed $\omega$ when removing the minimization operator. Based on this, we obtain the result of Eq. 12. In terms of the sufficient and necessary condition for equality, we note that for any optimal $\omega^*$ we have

$$\omega^* z^2 - \tilde{\rho}(\omega^*) = \rho(z^2), \tag{15}$$

which is tangent to $\rho$ at $z^2$ and $\omega^* = \frac{\partial \delta(z^2)}{\partial z^2}$. We thus obtain the result of Eq. 13. $\qquad\square$

We next continue to prove the main result of Theorem 1. According to Proposition 1, we can minimize the upper bound surrogate Eq. 12 and it becomes equivalent to a minimization of the original energy on condition that the variational parameters are given by Eq. 13. Then with a one-step gradient decent of Eq. 12, the instance states could be updated via (assuming $l$ as the index of steps and $\alpha$ as step size)

$$
\begin{aligned}
\mathbf{Z}^{(l+1)} &= \mathbf{Z}^{(l)} - \alpha \left.\frac{\partial \tilde{E}(\mathbf{Z};\delta)}{\partial \mathbf{Z}}\right|_{\mathbf{Z}=\mathbf{Z}^{(l)}} \\
&= \mathbf{Z}^{(l)} - \alpha \left(\lambda(\mathbf{D}^{(l)} - \mathbf{\Omega}^{(l)})\mathbf{Z}^{(l)} + \mathbf{Z}^{(l)} - \mathbf{Z}^{(l)}\right) \\
&= \mathbf{Z}^{(l)} - \alpha'(\mathbf{D}^{(l)} - \mathbf{\Omega}^{(l)})\mathbf{Z}^{(l)}
\end{aligned}
\tag{16}
$$

where $\mathbf{\Omega}^{(l)} = \{\omega_{ij}^{(l)}\}_{N \times N}$, $\mathbf{D}^{(l)}$ denotes the diagonal degree matrix associated with $\mathbf{\Omega}^{(l)}$ and we introduce $\alpha' = \alpha\lambda$ to combine two parameters as one. Common practice to accelerate convergence adopts a positive definite preconditioner term, e.g., $(\mathbf{D}^{(l)})^{-1}$, to re-scale the updating gradient and the final updating form becomes

$$\mathbf{Z}^{(l+1)} = (1 - \alpha')\mathbf{Z}^{(l)} + \alpha'(\mathbf{D}^{(l)})^{-1}\mathbf{\Omega}^{(l)}\mathbf{Z}^{(l)}. \tag{17}$$

One can notice that Eq. 17 shares similar forms as the numerical iteration Eq. 3 for the PDE diffusion system, in particular if we write Eq. 3 as a matrix form:

$$\mathbf{Z}^{(k+1)} = \left(1 - \tau\tilde{\mathbf{D}}^{(k)}\right)\mathbf{Z}^{(k)} + \tau\mathbf{S}^{(k)}\mathbf{Z}^{(k)}. \tag{18}$$

where $\tilde{\mathbf{D}}^{(k)}$ is the degree matrix associated with $\mathbf{S}^{(k)}$. Pushing further, we can see that the effect of Eq. 18 is the same as 17 when we let $\tau = \alpha'$, $k = l$, $\mathbf{S}^{(k)} = (\mathbf{D}^{(l)})^{-1}\mathbf{\Omega}^{(l)}$ and $\mathbf{S}^{(k)}$ is row-normalized, i.e., $\sum_{v \in V} \mathbf{S}_{uv}^{(k)} = 1$ and $\tilde{\mathbf{D}}^{(k)} = \mathbf{I}$.

Thereby, we have proven by construction that a one-step numerical iteration by the explicit Euler scheme, specifically shown by Eq. 17 is equivalent to a one-step gradient descent on the surrogate Eq. 14 which further equals to the original energy Eq. 4. We thus have the result $E(\mathbf{Z}^{(k+1)}, k; \delta) \leq E(\mathbf{Z}^{(k)}, k; \delta)$. Besides, we notice that for a fixed $\mathbf{Z}$, $E(\mathbf{Z}, k; \delta) = \|\mathbf{Z} - \mathbf{Z}^{(k)}\|_{\mathcal{F}}^2 + \lambda\sum_{ij}\rho(\|\mathbf{z}_i - \mathbf{z}_j\|_2^2)$ becomes a function of $k$ and its optimum is achieved if and only if $\mathbf{Z}^{(k)} = \mathbf{Z}$. Such a fact yields that $E(\mathbf{Z}^{(k)}, k; \delta) \leq E(\mathbf{Z}^{(k)}, k - 1; \delta)$. The result of the main theorem follows by noting that $E(\mathbf{Z}^{(k+1)}, k; \delta) \leq E(\mathbf{Z}^{(k)}, k; \delta) \leq E(\mathbf{Z}^{(k)}, k - 1; \delta)$.

## C  Extension of our Theory to Incorporate Non-Linearity

In Section 3, we mainly focus on the situation without non-linearity in each model layer to stay consistent with our implementation where we empirically found that omitting the non-linearity in the middle diffusion layers works smoothly in practice (see the pseudo code of Alg. 1 in appendix for details). Even so, our theory can also be extended to incorporate the layer-wise non-linearity in diffusion propagation. Specifically, the non-linear activation can be treated as a proximal operator (which projects the output into a feasible region) and the gradient descent used in our analysis Eqn. 16 and 17 can be modified to add a proximal operator:

$$\mathbf{Z}^{(l+1)} = \mathrm{Prox}_\Omega \left( (1 - \alpha)\mathbf{Z}^{(l)} + \alpha(\mathbf{D}^{(l)})^{-1}\mathbf{\Omega}^{(l)}\mathbf{Z}^{(l)} \right),$$

where $\mathrm{Prox}_\Omega(z) = \arg\min_{x \in \Omega} \|x - z\|_2$ and $\Omega$ defines a feasible region. The updating above corresponds to proximial gradient descent which also guarantees a strict minimization for the energy function and our Theorem 1 will still hold. In particular, if one uses ReLU activation, the proximal operator will be $\mathrm{Prox}_\Omega(z) = \max(0, z)$.

## D  Different Energy Forms

We present more detailed illustration for the choices of $f$ and specific energy function forms in Eq. 4.

**Simple Diffusivity Model.** As discussed in Section 4, the simple model assumes $f(z^2) = 2 - \frac{1}{2}z^2$ that corresponds to $g(x) = 1 + x$, where we define $z = \|\mathbf{z}_i - \mathbf{z}_j\|_2$ and $x = \mathbf{z}_i^\top \mathbf{z}_j$. The corresponding penalty function $\delta$ whose first-order derivative is $f$ would be $\delta(z^2) = 2z^2 - \frac{1}{4}z^4$. We plot the penalty function curves in Fig. 3(a). As we can see, the $f$ is a non-negative, decreasing function of $z^2$, which implies that the $\delta$ satisfies the non-decreasing and concavity properties to guarantee a valid regularized energy function. Also, in Fig. 3(a) we present the divergence field produced by 10 randomly generated instances (marked as red stars) and the cross-section energy field of one instance.

**Advanced Diffusivity Model.** The diffusivity model defines $f(z^2) = \frac{1}{1+e^{z^2/2-1}}$ with $g(x) = \frac{1}{1+\exp(-x)}$, and the corresponding penalty function $\delta(z^2) = z^2 - 2\log(e^{z^2/2-1} + 1)$. The penalty function curves, divergence field and energy field are shown in Fig. 3(b).

**Incorporating Input Graphs.** In Section 4, we present an extended version of our model for incorporating input graphs. Such an extension defines a new diffusion process whose iterations are equivalent (up to a re-scaling factor on the adjacency matrix) to a sequence of descending steps for the following regularized energy:

$$E(\mathbf{Z}, k; \delta) = \|\mathbf{Z} - \mathbf{Z}^{(k)}\|_\mathcal{F}^2 + \frac{\lambda}{2} \sum_{i,j} \delta(\|\mathbf{z}_i - \mathbf{z}_j\|_2^2) + \frac{\lambda}{2} \sum_{(i,j) \in \mathcal{E}} \|\mathbf{z}_i - \mathbf{z}_j\|_2^2, \tag{19}$$

where the last term contributes to a penalty for observed edges in the input graph.

## E  Model Implementation and Pseudo Codes

In this section, we present the details for model implementation including the feed-forward and loss computation of DIFFORMER.

**Input Layer.** For input data $\mathbf{X} = \{\mathbf{x}_i\}_{i=1}^N \in \mathbb{R}^{N \times D}$ where $\mathbf{x}_i$ denotes the $D$-dimensional input features of the $i$-th instance, we first use a shallow fully-connected layer to convert it into a $d$-dimensional embedding in the latent space:

$$\mathbf{Z} = \sigma\left(\mathrm{LayerNorm}(\mathbf{W}_I\mathbf{X} + \mathbf{b}_I)\right), \tag{20}$$

where $\mathbf{W}_I \in \mathbb{R}^{d \times D}$ and $\mathbf{b}_I \in \mathbb{R}^d$ are trainable parameters, and $\sigma$ is a non-linear activation (i.e., ReLU). Then the node embeddings $\mathbf{Z}$ will be used for feature propagation with our diffusion-induced Transformer model by letting $\mathbf{Z}^{(0)} = \mathbf{Z}$ as the initial states.

**Propagation Layer.** The updating rule for node embeddings at the $k$-th layer consists of a sequence of operations as described below. For the $h$-th head,

$$\mathbf{K}_h^{(k)} = \mathbf{W}_{K,h}^{(k)}\mathbf{Z}^{(k)}, \quad \mathbf{Q}_h^{(k)} = \mathbf{W}_{Q,h}^{(k)}\mathbf{Z}^{(k)}, \quad \mathbf{V}_h^{(k)} = \mathbf{W}_{V,h}^{(k)}\mathbf{Z}^{(k)}, \tag{21}$$

where $\mathbf{W}_K^{(k)}{}_h \in \mathbb{R}^{d \times d}, \mathbf{W}_Q^{(k)}{}_h \in \mathbb{R}^{d \times d}, \mathbf{W}_V^{(k)}{}_h \in \mathbb{R}^{d \times d}$ are trainable parameters of the $h$-th head at the $k$-th layer. Then the transformed embeddings will be fed into the all-pair propagation unit with different specifications for DIFFORMER-s and DIFFORMER-a.

- For DIFFORMER-s: we first adopt L2 normalization for key and query vectors

$$\tilde{\mathbf{K}}_h^{(k)} = \left[ \frac{\mathbf{K}_{h,i}^{(k)}}{\|\mathbf{K}_{h,i}^{(k)}\|_2} \right]_{i=1}^N, \quad \tilde{\mathbf{Q}}_h^{(k)} = \left[ \frac{\mathbf{Q}_{h,i}^{(k)}}{\|\mathbf{Q}_{h,i}^{(k)}\|_2} \right], \tag{22}$$

where $\mathbf{K}_{h,i}^{(k)}$ denotes the $i$-th row vector of $\mathbf{K}_h^{(k)}$. Then the all-pair propagation is achieved by

$$\mathbf{R}_h^{(k)} = \text{diag}^{-1} \left( N + \tilde{\mathbf{Q}}_h^{(k))} \left( (\tilde{\mathbf{K}}_h^{(k)})^\top \mathbf{1} \right) \right), \tag{23}$$

$$\mathbf{P}_h^{(k)} = \mathbf{R}_h^{(k)} \left[ N \cdot \mathbf{V}_h^{(k)} + \tilde{\mathbf{Q}}_h^{(k)} \left( (\tilde{\mathbf{K}}_h^{(k)})^\top \mathbf{V}^{(k)} \right) \right]. \tag{24}$$

The above computation only requires $O(N)$ complexity.

- For DIFFORMER-a: we need to compute tha all-pair similarity before aggregating the results

$$\mathbf{A}_h^{(k)} = \text{Sigmoid} \left( \mathbf{Q}_h^{(k)} (\mathbf{K}_h^{(k)})^\top \right), \tag{25}$$

$$\mathbf{R}_h^{(k)} = \text{diag}^{-1} \left( \mathbf{A}_h^{(k)} \mathbf{1} \right), \tag{26}$$

$$\mathbf{P}_h^{(k)} = \mathbf{R}_h^{(k)} \mathbf{A}_h^{(k)}. \tag{27}$$

If using input graphs, we add the updated embeddings of graph-based propagation to the all-pair propagation's ones:

$$\overline{\mathbf{P}}_h^{(k)} = \mathbf{P}_h^{(k)} + \mathbf{D}^{-\frac{1}{2}} \mathbf{A} \mathbf{D}^{-\frac{1}{2}} \mathbf{V}_h^{(k)}, \tag{28}$$

where $\mathbf{A}$ is the input graph and $\mathbf{D}$ denotes its corresponding diagonal degree matrix.

We then average the propagated results of multiple heads:

$$\overline{\mathbf{P}}^{(k)} = \frac{1}{H} \sum_{h=1}^H \overline{\mathbf{P}}_h^{(k)}. \tag{29}$$

After once propagation, the next-layer embeddings will be updated by

$$\mathbf{Z}^{(k+1)} = \sigma' \left( \text{LayerNorm} \left( \tau \overline{\mathbf{P}}^{(k)} + (1 - \tau) \mathbf{Z}^{(k)} \right) \right), \tag{30}$$

where $\sigma'$ can be identity mapping or non-linear activation (e.g., ReLU).

**Output Layer.** After $K$ layers of propagation, we then use a shallow fully-connected layer to output the predicted logits:

$$\hat{\mathbf{Y}} = \mathbf{Z}^{(K)} \mathbf{W}_O + \mathbf{b}_O, \tag{31}$$

where $\mathbf{W}_O \in \mathbb{R}^{d \times C}$ and $\mathbf{b}_O \in \mathbb{R}^C$ are trainable parameters, and $C$ denotes the number of classes. And, the predicted logits $\hat{\mathbf{Y}}$ will be used for computing a loss of the form $l(\hat{\mathbf{Y}}, \mathbf{Y})$ where $l$ can be cross-entropy for classification or mean square error for regression.

### E.1 PSEUDO CODES

We provide the Pytorch-style pseudo codes for DIFFORMER class in Alg. 1 and the one-layer propagation of two model versions (shown in Alg. 2 for DIFFORMER-s and Alg. 3 for DIFFORMER-a). The key design of our methodology lies in the model architectures which are shown in detail in Alg. 2 for DIFFORMER-s and Alg. 3 for DIFFORMER-a, where for each case, the model takes the data as input and outputs prediction for each individual instance. For more details concerning the implementation, please refer to our provided codes at the first page.

**Algorithm 1** PyTorch-style Code for DIFFORMER

```
# fcs: fully-connected layers
# bns: layer normalization layers
# convs: DIFFormer layers (see implementation in Alg. 2 and 3)
# activation: activation function for the input layer
# x: input data sized [N, D], N for instance number, D for input feature dimension
# edge_index: input graph structure if available, None otherwise
# tau: step size for each iteration update
# use_act: whether to use activation for propagation layers

layer_ = []

# input MLP layer
x = fcs[0](x)
x = bns[0](x)
x = activation(x)

# store as residual link
layer_.append(x)

for i, conv in enumerate(convs):
    # graph convolution with global all-pair attention (specified by Alg. 2 for DIFFormer
        -s and Alg. 3 for DIFFormer-a)
    x = conv(x, x, edge_index)
    x = tau * x + (1-tau) * layer_[i]
    x = bns[i+1](x)
    if use_act:
        x = activation(x)
    layer_.append(x)

# output MLP layer
out = fcs[-1](x)

# supervised loss calculation, negative log-likelihood
y_logp = F.log_softmax(out, dim=1)
loss = criterion(y_logp[train_idx], y_true[train_idx])
```

**Algorithm 2** PyTorch-style Code for One-layer Feed-forward of DIFFORMER-s

```
# x: data embeddings sized [N, d], N for instance number, d for hidden size
# edge_index: input graph structure if available, None otherwise
# H: head number
# use_graph: whether to use input graph
# use_weight: whether to use feature transformation for each layer
# graph_conv: graph convolution operatior using the normalized adjacency matrix D^{-1/2}
    AD^{-1/2}
# Wq, Wk, Wv: weight matrices for feature transformation

Q = Wq(x) # [N, H, D]
K = Wk(x) # [N, H, D]
V = use_weight * Wv(x) + (1 - use_weight) * x # [N, H, D]

# numerator
KV = torch.einsum("lhm,lhd->hmd", K, V)
num = torch.einsum("nhm,hmd->nhd", Q, KV) # [N, H, D]
num += N * V

# denominator
all_ones = torch.ones(N)
K_sum = torch.einsum("lhm,l->hm", K, all_ones)
den = torch.einsum("nhm,hm->nh", Q, K_sum) # [N, H]

# aggregated results
den += torch.ones_like(den) * N
agg = num / den.unsqueeze(2) # [N, H, D]

# use input graph for graph conv
if use_graph:
    agg = agg + graph_conv(V, edge_index)
output = agg.mean(dim=1)
```

**Algorithm 3** PyTorch-style Code for One-layer Feed-forward of DIFFORMER-a

```
# x: data embeddings sized [N, d], N for instance number, d for hidden size
# edge_index: input graph structure if available, None otherwise
# H: head number
# use_graph: whether to use input graph
# use_weight: whether to use feature transformation for each layer
# graph_conv: graph convolution operatior using the normalized adjacency matrix D^{-1/2}
    AD^{-1/2}
# Wq, Wk, Wv: weight matrices for feature transformation

Q = Wq(x)  # [N, H, D]
K = Wk(x)  # [N, H, D]
V = use_weight * Wv(x) + (1 - use_weight) * x # [N, H, D]

# numerator
num = torch.sigmoid(torch.einsum("nhm,lhm->nlh", Q, K)) # [N, N, H]

# denominator
all_ones = torch.ones(N)
den = torch.einsum("nlh,l->nh", num, all_ones)
den = den.unsqueeze(1).repeat(1, N, 1) # [N, N, H]

# aggregated results
attn = num / den # [N, N, H]
agg = torch.einsum("nlh,lhd->nhd", attn, V) # [N, H, D]

# use input graph for graph conv
if use_graph:
    agg = agg + graph_conv(V, edge_index)
output = agg.mean(dim=1)
```

# F    CONNECTIONS WITH EXISTING MODELS

**MLP.** MLPs can be viewed as a simplified diffusion model with only non-zero diffusivity values on the diagonal line, i.e., $\mathbf{S}_{ij}^{(k)} = 1$ if $i = j$ and 0 otherwise. From a graph convolution perspective, MLP only considers propagation on the self-loop connection in each layer. Correspondlying, the energy function the feed-forward process of the model essentially optimizes would be $E(\mathbf{Z}, k) = \|\mathbf{Z} - \mathbf{Z}^{(k)}\|_{\mathcal{F}}^2$, which only counts for the local consistency term and ignores the global information.

**GCN.** Graph Convolution Networks (Kipf & Welling, 2017) define a convolution operator on a graph $\mathcal{G} = (\mathcal{V}, \mathcal{E})$ by multiplying the node feature matrix with $D^{-\frac{1}{2}}AD^{-\frac{1}{2}}$ where $A$ denotes the adjacency matrix and $D$ is its associated degree matrix. The layer-wise updating rule can be written as

$$
\hat{\mathbf{S}}_{ij}^{(k)} = \begin{cases} \dfrac{1}{\sqrt{d_i d_j}}, & \text{if } (i,j) \in \mathcal{E}, \\ 0, & otherwise, \end{cases}
$$
$$
\mathbf{z}_i^{(k+1)} = \left(1 - \tau \sum_{j=1}^{N} \hat{\mathbf{S}}_{ij}^{(k)}\right) \mathbf{z}_i^{(k)} + \tau \sum_{j=1}^{N} \hat{\mathbf{S}}_{ij}^{(k)} \mathbf{z}_j^{(k)}, \quad 1 \le i \le N. \tag{32}
$$

Eq. 32 generalizes the original message-passing rule of GCN by adding an additional self-loop links with adaptive weights for different nodes. The diffusivity matrix is defined with the observed adjacency, i.e., $\hat{\mathbf{S}}^{(k)} = D^{-\frac{1}{2}}AD^{-\frac{1}{2}}$. Though such a design leverages the geometric information from input structures as a guidance for feature propagation, it constrains the efficiency of layer-wise information flows within the receptive field of local neighbors and could only exploit partial global information with a finite number of iterations.

**GAT.** Graph Attention Networks (Velickovic et al., 2018) extend the GCN architecture to incorporate attention mechanisms as a learnable function producing adaptive weights for each observed edge. From our diffusion perspective, the attention matrix for layer-wise convolution can be treated as the

diffusivity in our updating rule:

$$\hat{\mathbf{S}}_{ij}^{(k)} = \frac{f(\|\mathbf{z}_i^{(k)} - \mathbf{z}_j^{(k)}\|_2^2)}{\sum_{(i,l) \in \mathcal{E}} f(\|\mathbf{z}_i^{(k)} - \mathbf{z}_l^{(k)}\|_2^2)}, \quad (i,j) \in \mathcal{E},$$

$$\mathbf{z}_i^{(k+1)} = \left(1 - \tau \sum_{j=1}^{N} \hat{\mathbf{S}}_{ij}^{(k)}\right) \mathbf{z}_i^{(k)} + \tau \sum_{j=1}^{N} \hat{\mathbf{S}}_{ij}^{(k)} \mathbf{z}_j^{(k)}, \quad 1 \le i \le N. \tag{33}$$

We notice that $\sum_{j=1}^{N} \hat{\mathbf{S}}_{ij}^{(k)} = 1$ due to the normalization in the denominator. Therefore, the layer-wise updating can be viewed as an attentive aggregation over graph structures and a subsequent residual link. In fact, the original implementation for the GAT model (Velickovic et al., 2018) specifies the function $f$ as a particular form, i.e., $\exp(\text{LeakyReLU}(\mathbf{a}^\top[\mathbf{W}\mathbf{z}_i^{(k)} \| \mathbf{W}\mathbf{z}_j^{(k)}]))$, which can be viewed as a generalized similarity function given trainable $\mathbf{W}$, $\mathbf{a}$ towards optimizing a supervised loss.

## G  DATASET INFORMATION

In this section, we present the detailed information for all the experimental datasets, the pre-processing and evaluation protocol used in Section 5.

Table 6: Information for node classification datasets.

| Dataset | Type | # Nodes | # Edges | # Node features | # Class |
|---|---|---|---|---|---|
| Cora | Citation network | 2,708 | 5,429 | 1,433 | 7 |
| Citeseer | Citation network | 3,327 | 4,732 | 3,703 | 6 |
| Pubmed | Citation network | 19,717 | 44,338 | 500 | 3 |
| Proteins | Protein interaction | 132,534 | 39,561,252 | 8 | 2 |
| Pokec | Social network | 1,632,803 | 30,622,564 | 65 | 2 |

### G.1  NODE CLASSIFICATION DATASETS

`Cora`, `Citeseer` and `Pubmed` (Sen et al., 2008) are commonly used citation networks for evaluating models on node classification tasks., These datasets are small-scale networks (with 2K~20K nodes) and the goal is to classify the topics of documents (instances) based on input features of each instance (bag-of-words representation of documents) and graph structure (citation links). Following the semi-supervised learning setting in Kipf & Welling (2017), we randomly choosing 20 instances per class for training, 500/1000 instances for validation/testing for each dataset. `OGBN-Proteins` (Hu et al., 2020) is a multi-task protein-protein interaction network whose goal is to predict molecule instances' property. We follow the original splitting of Hu et al. (2020) for evaluation. `Pokec` is a large-scale social network with features including profile information, such as geographical region, registration time, and age, for prediction on users' gender. For semi-supervised learning, we consider randomly splitting the instances into train/valid/test with 10%/10%/80% ratios. Table 6 summarizes the statistics of these datasets.

### G.2  IMAGE AND TEXT CLASSIFICATION DATASETS

We evaluate our model on two image classification datasets: STL-10 and CIFAR-10. We use all 13000 images from STL-10, each of which belongs to one of ten classes. We choose 1500 images from each of 10 classes of CIFAR-10 and obtain a total of 15,000 images. For STL-10 and CIFAR-10, we randomly select 10/50/100 instances per class as training set, 1000 instances for validation and the remaining instances for testing. We first use the self-supervised approach SimCLR (Chen et al., 2020b) (that does not use labels for training) to train a ResNet-18 for extracting the feature maps as input features of instances. We also evaluate our model on 20Newsgroup, which is a text classification dataset consisting of 9607 instances. We follow Franceschi et al. (2019) to take 10 classes from 20 Newsgroup and use words (TFIDF) with a frequency of more than 5% as features.

### G.3 Spatial-Temporal Datasets

The spatial-temporal datasets are from the open-source library PyTorch Geometric Temporal (Rozemberczki et al., 2021), with properties and summary statistics described in Table 7. Node features are evolving for all the datasets considered here, i.e., we have different node features for different snapshots. For each dataset, we split the snapshots into training, validation, and test sets according to a 2:2:6 ratio in order to make it more challenging and close to the real-world low-data learning setting. In details:

- `Chickenpox` describes weekly officially reported cases of chickenpox in Hungary from 2005 to 2015, whose nodes are counties and edges denote direct neighborhood relationships. Node features are lagged weekly counts of the chickenpox cases (we included 4 lags). The target is the weekly number of cases for the upcoming week (signed integers).

- `Covid` contains daily mobility graph between regions in England NUTS3 regions, with node features corresponding to the number of confirmed COVID-19 cases in the previous days from March to May 2020. The graph indicates how many people moved from one region to the other each day, based on Facebook Data For Good disease prevention maps. Node features correspond to the number of COVID-19 cases in the region in the past 8 days. The task is to predict the number of cases in each node after 1 day.

- `WikiMath` is a dataset whose nodes describe Wikipedia pages on popular mathematics topics and edges denote the links from one page to another. Node features are provided by the number of daily visits between 2019 March and 2021 March. The graph is directed and weighted. Weights represent the number of links found at the source Wikipedia page linking to the target Wikipedia page. The target is the daily user visits to the Wikipedia pages between March 16[th] 2019 and March 15[th] 2021 which results in 731 periods.

Table 7: Properties and summary statistics of the spatial-temporal datasets used in the experiments with information about whether the graph structure is dynamic or static, meaning of node features (the same as the prediction target) and the corresponding dimension ($D$), the number of snapshots ($T$), the number of nodes ($|V|$), as well as the meaning of edges/edge weights.

| Dataset | Graph structure | Node features/ Target | $D$ | Frequency | $T$ | $|V|$ | Edges/ Edge weights |
|---------|----------------|----------------------|-----|-----------|-----|-------|---------------------|
| Chickenpox | Static | Weekly Chickenpox Cases | 4 | Weekly | 522 | 20 | Direct Neighborhoods |
| Covid | Dynamic | Daily Covid Cases | 8 | Daily | 61 | 129 | Daily Mobility |
| WikiMath | Static | Daily User Visits | 14 | Daily | 731 | 1,068 | Page Links |

## H  Implementation Details and Hyper-parameters

### H.1  Node Classification Experiment

We use feature transformation for each layer on two large datasets and omit it for citation networks. The head number is set as 1. We set $\tau = 0.5$ and incorporate the input graphs for DIFFORMER. For other hyper-paramters, we adopt grid search for all the models with learning rate from $\{0.0001, 0.001, 0.01, 0.1\}$, weight decay for the Adam optimizer from $\{0, 0.0001, 0.001, 0.01, 0.1, 1.0\}$, dropout rate from $\{0, 0.2, 0.5\}$, hidden size from $\{16, 32, 64\}$, number of layers from $\{2, 4, 8, 16\}$. For evaluation, we compute the mean and standard deviation of the results with five repeating runs with different initializations. For each run, we run for a maximum of 1000 epochs and report the testing performance achieved by the epoch yielding the best performance on validation set.

### H.2  Image and Text Classification Experiment

For image and text datasets, we consider feature transformation for layer-wise updating. The head number is set as 1. We set $\tau = 0.5$. These datasets do not have input graphs so we only consider learning new structures for the diffusion model. For hyper-parameter settings, we conduct grid search for all the models with learning rate from $\{0.0001, 0.0005, 0.005, 0.01, 0.05\}$, weight decay for the Adam optimizer from $\{0.0001, 0.001, 0.01, 0.1\}$, dropout rate from $\{0, 0.2, 0.5\}$, hidden size from

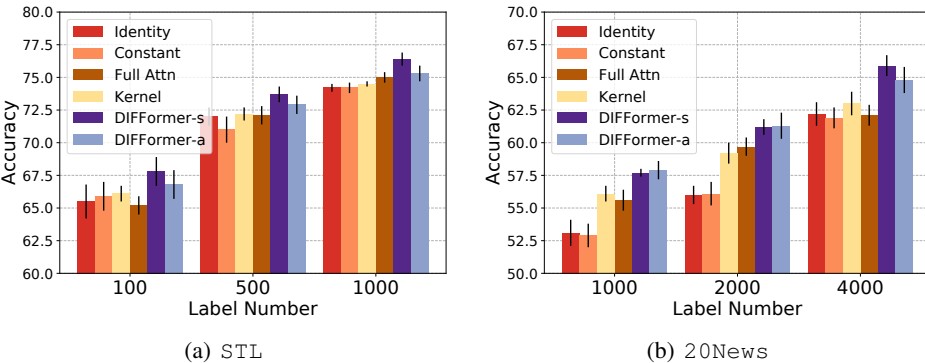

Figure 4: Ablation studies for different energy function forms on image and text datasets.

$\{32, 64, 100, 200, 300, 400\}$, number of layers from $\{1, 2, 4, 6, 8, 10, 12\}$. We average the results for five repeating runs and report as well the standard deviation. For each run, we run for a maximum of 600 epochs and report the testing accuracy achieved by the epoch yielding the highest accuracy on validation set.

### H.3 SPATIAL-TEMPORAL PREDICTION EXPERIMENT

We do not use feature transformation for these datasets due to their small sizes and also set $\tau = 0.5$. The head number is set as 1. These spatial-temporal dynamics prediction datasets contain available graph structures, we consider both cases, using the input graphs and not, in our experiments and discuss their impact on the performance. For other hyper-parameters, we also consider grid search for all models here with learning rate from $\{0.01, 0.05, 0.005\}$, weight decay for the Adam optimizer from $\{0, 0.005\}$, dropout rate from $\{0, 0.2, 0.5\}$, and report the test mean squared error (MSE) based on the lowest validation MSE. We average the results for five repeating runs and report as well the standard deviation for each MSE result. For each run, we run for a maximum of 200 epochs in total and stop the training process with 20-epoch early stopping on the validation performance. The data split is done in time order, and hence is deterministic. We report the results using the same hidden size (4) and number of layers (2) for all methods.

## I MORE EXPERIMENT RESULTS

We supplement more experiment results including extra ablation studies, hyper-parameter studies and visualization results on more datasets that are not presented in Section 5 due to the limit of space.

### I.1 ABLATION STUDIES

In Fig. 4 we present more experiment results for ablation studies w.r.t. the energy function forms used by DIFFORMER. See discussions and analysis in Section 5.

### I.2 HYPER-PARAMETER ANALYSIS

We plot the testing performance of several baselines and DIFFORMER with different step size $\tau$ as the model size $K$ increases in Fig. 5. See discussions and analysis in Section 5.

### I.3 VISUALIZATION

Fig. 6 and 7 plot the produced instance-level representations and diffusion strength estimates by the model on 20News and STL, respectively. We observe that the diffusivity estimates tend to connect nodes with different classes, which contribute to increasing the global connectivity and facilitate absorbing other instances' information for informative representations. The node embeddings

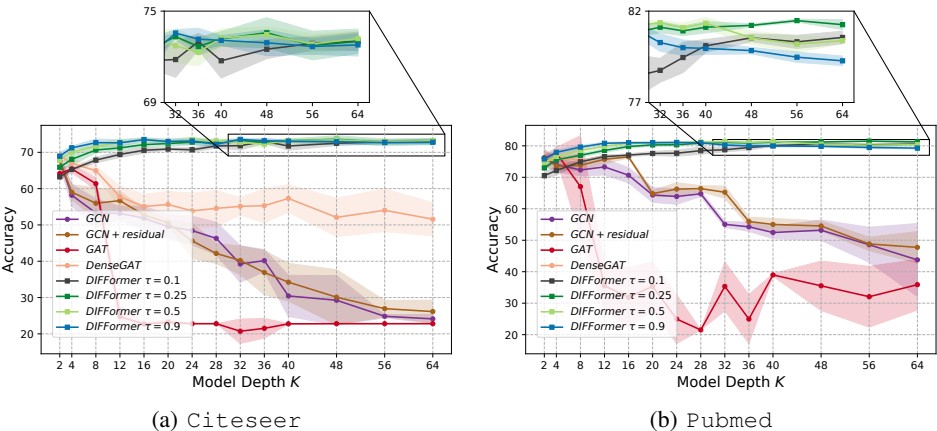

(a) Citeseer                          (b) Pubmed

Figure 5: Hyper-parameter studies of model depth $K$ and step size $\tau$ on two citation networks.

produced by our model have small intra-class distance and large inter-class distance, making it easier for the classifier to distinguish.

Fig. 8 visualizes the diffusivity estimates on Chickenpox. We conclude that large diffusion strength usually exists between nodes with similar ground-truth labels. DIFFORMER-s has more concentrated large weights while DIFFORMER-a tends to have large diffusivity spreading out more. DIFFORMER-a indeed learns more complex underlying structures than DIFFORMER-s due to its better capacity for diffusivity modeling.

## I.4 IMPACT OF MINI-BATCH SIZES ON LARGE GRAPHS

The randomness of mini-batch partition on large graphs has negligible effect on the performance since we use large batch sizes for training, which is facilitated by the linear complexity of DIFFORMER-s. Even setting the batch size to be 100000, our model only costs 3GB GPU memory on Pokec. As a further investigation on this, we add more experiments using different batch sizes on Pokec and the results are shown in Table 8.

Table 8: Discussions on using different mini-batch sizes for training on Pokec. We report testing accuracy and training memory for comparison.

| Batch size | 5000 | 10000 | 20000 | 50000 | 100000 | 200000 |
|---|---|---|---|---|---|---|
| Test Acc (%) | $65.24 \pm 0.34$ | $67.48 \pm 0.81$ | $68.53 \pm 0.75$ | $68.96 \pm 0.63$ | $69.24 \pm 0.76$ | $69.15 \pm 0.52$ |
| GPU Memory (MB) | 1244 | 1326 | 1539 | 2060 | 2928 | 4011 |

One can see that using small batch sizes would indeed sacrifice the performance yet large batch sizes can produce decent and low-variance results with acceptable memory costs.

## I.5 COMPARISON OF RUNNING TIME AND MEMORY COSTS

To further study the efficiency and scalability of our model, we provide more comparison regarding the training time per epoch and memory costs of two DIFFORMER's variants, GCN, GAT and DenseGAT in Table 9. One can see that compared to GAT, DIFFORMER-s costs comparable time on small datasets such as Cora and WikiMath, and is much faster on large dataset Pokec. As for memory consumption, DIFFORMER-s reduces the costs by several times over DenseGAT, which clearly shows the efficiency of our new diffusion function designs. Overall, DIFFORMER-s has nice scalability, decent efficiency and yield significantly better accuracy. In contrast, DIFFORMER-a costs much larger time and memory costs than DIFFORMER-s, due to its quadratic complexity induced by the explicit computation for the all-pair diffusivity. Still, DIFFORMER-a accommodates non-linearity for modeling the diffusion strengths which enables better capacity for learning complex layer-wise inter-interactions.

Table 9: Comparison of training time and memory of different models on `Cora`, `Pokec`, `STL-10` and `WikiMath`. OOM refers to out-of-memory when training on a GPU with 16GB memory.

| | Method | GCN | GAT | DenseGAT | DIFFORMER-s | DIFFORMER-a |
|---|---|---|---|---|---|---|
| Cora | Train time (s) | 0.0584 | 0.0807 | 0.5165 | 0.1438 | 0.3292 |
| | Training memory (MB) | 1168 | 1380 | 8460 | 1350 | 3893 |
| Pokec | Train time (s) | 1.069 | 14.87 | 88.07 | 2.206 | OOM |
| | Training memory (MB) | 1812 | 2014 | 13174 | 2923 | OOM |
| STL | Train time (s) | 0.0069 | 0.0424 | OOM | 0.0323 | 0.3298 |
| | Training memory (MB) | 1224 | 1980 | OOM | 1342 | 7680 |
| WikiMath | Train time (s) | 0.0081 | 0.0261 | 0.0364 | 0.0281 | 0.0350 |
| | Training memory (MB) | 1048 | 1054 | 1316 | 1046 | 1142 |

## I.6 INCORPORATION OF PSEUDO LABELS

For semi-supervised learning, there is a line of approaches that leverage pseudo labels to augment the training data. Our model DIFFORMER essentially has orthogonal technical aspects compared to this line of work in that we focus on building a new encoder backbone and only train the model with a standard supervised loss on the labeled data. This means that pseudo-label-based approaches are equally applicable for enhancing the training of our model as well as the competitors we used in our experiments.

As an initial verification of this claim, we use the Meta Pseudo Labels (MPL) (Pham et al., 2021) as a plug-in module to boost DIFFORMER as well as our competitors GCN-kNN and GAT-kNN, and empirically compare the relative improvement. Specifically, we use DIFFORMER-s, DIFFORMER-a, GCN and GAT as the encoder backbone of teacher and student models, respectively, and use the MPL algorithm to generate pseudo labels for augmenting the training data used for computing the supervised loss. The results on `CIFAR-10` and `STL-10` are shown in Table 10. As we can see, the MPL contributes to some performance gains across all four encoders, while our two DIFFORMER variants still maintain the superiority over the competitors. Note also that as a proof-of-concept here we did not use an additional consistency loss that requires careful manual tuning. However, in practice this type of more sophisticated MPL implementation could in principle be applied to further improve performance (across all models).

Table 10: Comparison of using and not using Meta Pseudo Labels (MPL) as a plug-in module to boost different encoder backbones on `CIFAR-10` and `STL-10`.

| | Method | GCN | GAT | DIFFORMER-s | DIFFORMER-a |
|---|---|---|---|---|---|
| STL | w/o MPL | $73.7 \pm 0.4$ | $73.9 \pm 0.6$ | $76.4 \pm 0.5$ | $75.3 \pm 0.6$ |
| | w/ MPL | $74.3 \pm 0.5$ | $74.5 \pm 0.7$ | $77.0 \pm 0.6$ | $75.9 \pm 0.4$ |
| CIFAR | w/o MPL | $74.7 \pm 0.5$ | $74.1 \pm 0.5$ | $76.6 \pm 0.3$ | $75.9 \pm 0.3$ |
| | w/ MPL | $75.3 \pm 0.4$ | $74.8 \pm 0.5$ | $77.1 \pm 0.3$ | $76.3 \pm 0.3$ |

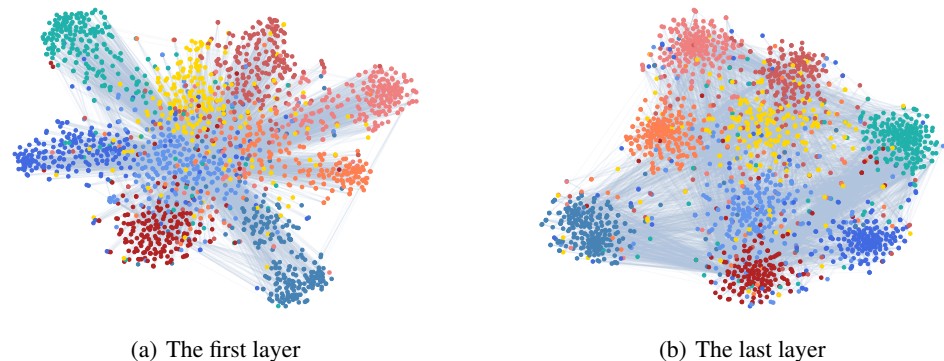

(a) The first layer            (b) The last layer

Figure 6: Visualization of instance representations and diffusivity strengths (we set a threshold and only plot the edges with weights more than the threshold) at different layers given by DIFFORMER-s on `20News`.

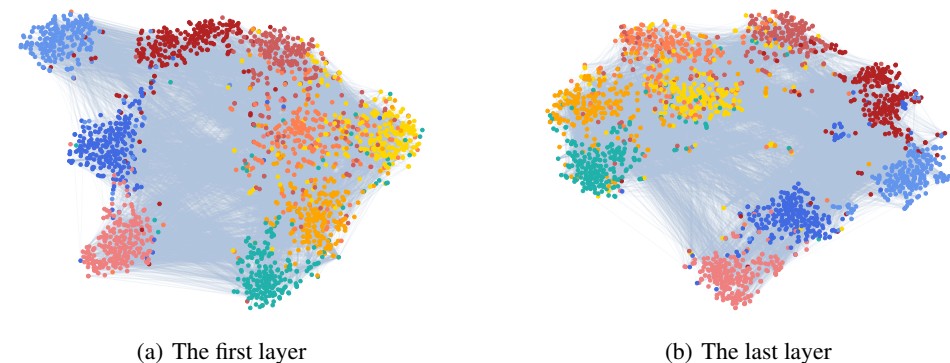

(a) The first layer            (b) The last layer

Figure 7: Visualization of instance representations and diffusivity strengths (we set a threshold and only plot the edges with weights more than the threshold) at different layers given by DIFFORMER-s on `STL`.

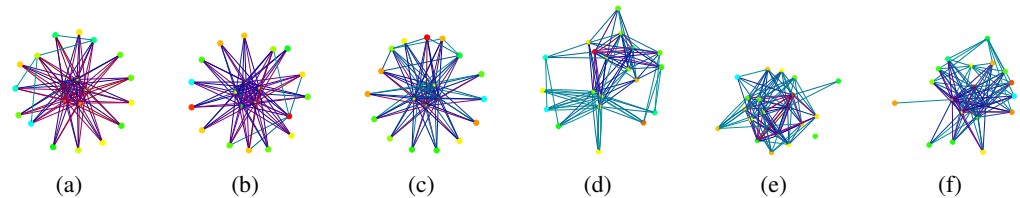

(a)     (b)     (c)     (d)     (e)     (f)

Figure 8: The produced diffusivity of the first layer (i.e., $\hat{\mathbf{S}}^{(1)}$) on `Chickenpox` across the first three snapshots, yielded by DIFFORMER-s, shown in (a)∼(c), and DIFFORMER-a, shown in (d)∼(f). Node colors correspond to ground-truth labels (i.e., reported cases), varying from red to blue as the label increases. We visualize the edges with top 100 diffusion strength, where edge colors change from blue to red as $\hat{\mathbf{S}}_{ij}^{(1)}$ increases.

