# OpenReview forum: "DIFFormer: Scalable (Graph) Transformers Induced by Energy Constrained Diffusion"
_ICLR.cc/2023/Conference — ICLR 2023 notable top 25%_

### Official Review · Reviewer_ckRq · 2022-10-23

**Confidence:** 3
**Correctness:** 4
**Technical Novelty And Significance:** 4
**Empirical Novelty And Significance:** 4
**Recommendation:** 8

**Clarity, Quality, Novelty And Reproducibility:**

- The paper is well-written and easy to follow. The approach is novel to the extent of the reviewer’s knowledge.
- The authors provided code and detailed instruction to reproduce the results.


**Strength And Weaknesses:**

Strength:
- This paper does a good job in motivating and describing the approach. From my perspective, the theory is quite interesting and elegant. The authors also provide a unified view for MLP, GCN, and GAT from their framework.
- The model achieves improved performance over a broad range of tasks.
- Extensive experiment results are provided in the Appendix on the influence of batch sizes, compute cost, etc.

Weakness:
- The model may have challenges in scaling to larger dataset due to the O(NKd^2) scaling for DIFFormer-s and O(N^2Kd^2) scaling for DIFFormer-a. I don’t fully understand the GPU memory cost result in Table 8. Why does the memory cost only increase by 4 times when the batch size increases from 5000 to 200000? How about the compute and memory cost for DIFFormer-a?
- For the CIFAR-10 and STL-10 experiments, have the authors considered stronger semi-supervised learning baselines like Meta Pseudo Labels?


**Summary Of The Paper:**

This paper introduces a new class of self-supervised encoder that learns the latent representation of data instances via a diffusion process that encodes the interaction interaction strength between data pairs. Two versions of the model are introduced, DIFFormer-s and DIFFormer-a, both demonstrating superior performance in a range of tasks.

**Summary Of The Review:**

In summary, this paper provides a novel, principled approach to learn the latent geometry of a dataset via an energy function that encodes pairwise diffusion strength. The theory is sound and well-motivated. Empirical results show improvements on a range of tasks.

---

> ### Author Response · Authors · 2022-11-14
> **Response to Reviewer ckRq**
>
> Thanks for the positive comments on our technical novelty, significance, and extensive experiments.
>
> **Q1: Scaling to larger datasets**
>
> The total complexity $O(NKd^2)$ for DIFFormer-s is desirable for scaling to large datasets, as shown by our experiments on data with up to 2M nodes/instances. We can also accommodate even larger datasets by following what we have done in Sec 5.2 using mini-batch training.  In this setting, using a large batch size of 0.1M only requires 3GB memory thanks to the linear complexity of DIFFormer-s w.r.t. instance number $N$. As for the $Kd^2$ term, which is comparable to other efficient models such as MLPs, we can hold $d$ (the hidden size) to a moderate number in practice for reducing the model size. Unlike DIFFormer-s, our variant DIFFormer-a has $O(N^2Kd^2)$ complexity and is not easily scalable to large batch sizes.
>
> **Q2: Memory cost w.r.t. batch size**
>
> There are many factors that influence the practical memory cost, and batch size is only one of them. To be more specific, on Pokec we also consider input graph structures for layer-wise propagation, so the graph adjacency (among nodes in one mini-batch) would require additional memory dependent on both the batch size and the edge density among nodes within each mini-batch. Hence the variation trend of memory costs may not strictly follow a steady increase as the batch size increases.
>
> **Q3: Compute and memory cost of DIFFormer-a**
>
> We have added more comparisons of the time/memory costs of DIFFormer-a in Appendix E.5. DIFFormer-a indeed costs much more time and memory due to its quadratic complexity induced by the explicit computation of all-pair diffusivity.
>
> **Q4: Discussion on using pseudo labels**
>
> Discussion of pseudo-labeling methods is a good suggestion that helps to showcase pathways for further improving our approach as follows. In this regard, Meta Pseudo Labeling (MPL) belongs to a line of semi-supervised learning approaches that leverage pseudo labels to augment the training data. Our model DIFFormer has orthogonal technical aspects compared to this line of work in that we focus on building a new encoder backbone and only train the model with a standard supervised loss on the labeled data. This means that MPL-based approaches are equally applicable for enhancing the training of our model as well as the competitors we used in our experiments.
>
> As an initial verification of this claim, we use MPL as a plug-in module to boost DIFFormer as well as our competitors GCN-knn and GAT-knn, and empirically compare the relative improvement. Speficially, we use DIFFormer-s, DIFFormer-a, GCN and GAT as the encoder backbone of teacher and student models, respectively, and use the MPL algorithm to generate pseudo labels for augmenting the training data used for computing the supervised loss. The results on CIFAR-10 and STL-10 are shown below:
>
> | STL | GCN-kNN | GAT-kNN | DIFFormer-s | DIFFormer-a
> | -------- | -------- | -------- | -------- | -------- |
> | w/o MPL | $73.7 \pm 0.4$ | $73.9 \pm 0.6$ | $76.4 \pm 0.5$ | $75.3 \pm 0.6$ |
> | w/ MPL | $74.3 \pm 0.5$ | $74.5 \pm 0.7$ | $77.0 \pm 0.6$ | $75.9 \pm 0.4$ |
>
> | CIFAR | GCN-kNN | GAT-kNN | DIFFormer-s | DIFFormer-a
> | -------- | -------- | -------- | -------- | -------- |
> | w/o MPL | $74.7 \pm 0.5$ | $74.1 \pm 0.5$ | $76.6 \pm 0.3$ | $75.9 \pm 0.3$ |
> | w/ MPL | $75.3 \pm 0.4$ | $74.8 \pm 0.5$ | $77.1 \pm 0.3$ | $76.3 \pm 0.3$ | |
>
>  As we can see, the MPL contributes to some performance gains across all four encoders, while our two DIFFormer variants still maintain the superiority over the competitors. We have added these results to the new Sec. H.6 in the Appendix. Note also that as a proof-of-concept here we did not use an additional consistency loss that requires careful manual tuning.  However, in practice this type of more sophisticated MPL implementation could in principle be applied to further improve performance (across all models).

---

### Official Review · Reviewer_cBoH · 2022-10-24

**Confidence:** 2
**Correctness:** 3
**Technical Novelty And Significance:** 3
**Empirical Novelty And Significance:** 3
**Recommendation:** 6

**Clarity, Quality, Novelty And Reproducibility:**

This is both a good discussion of a class of diffusion models which include and generalize many GNN models, and an empirical study of their use in graph learning.  Similar models are studied in physics and as a mathematical physicist I did not find the novelties claimed at the top of page 3 (the model definition and theorem 1) surprising.  I am not conversant enough with the ML literature to know how novel they are in that community.  Similarly the experimental results are nice but not so much better than previous quoted work to consider this a signficant advance.  Publishable but not obviously at the standard of this conference.

**Strength And Weaknesses:**

Strengths: Clearly written and many experimental results.

Weaknesses: This is a much studied field and I did not find the discussion of advances over previous work entirely convincing.

The performance improvements are marginal (1 sigma).

It was not clear to me how the "simple diffusivity model" could satisfy the conditions of theorem 1.  If the function g is linear in z^2, then \delta is quadratic.  But there is no nondecreasing concave quadratic function on R.  The "advanced diffusivity model" uses a nonpolynomial function.  Perhaps there is some implicit restriction on the domain, as suggested by the comment below Eq. (8) that f(z^2)=2-z^2/2 is non-negative, and the comment about layer normalization.  It is important to clarify this point.

Theorem 1 might follow from quasiconvexity of the energy function.

**Summary Of The Paper:**

In this work, a discrete diffusion process along the lines of Zhou et al (2004) but with features inspired by GNN works such as Velikovic et al (2017) is studied for use in semi-supervised node feature prediction.  It is argued that the process can be derived from an energy function.
Many experiments and comparisons with other models are done, and the results are competitive and arguably better.

**Summary Of The Review:**

Interesting contribution to a much studied field which makes incremental progress.

---

> ### Author Response · Authors · 2022-11-14
> **Response to Reviewer cBoH**
>
> Thanks for the valuable comments. We hope the following responses will serve to resolve any lingering reservations.
>
> **Q1: Limited significance of performance improvements**
>
> There are multiple reasons why we believe the performance improvements of our method are actually quite significant relative to existing methods.  First, we consistently obtain a roughly 1 sigma improvement over the best competitors across eleven datasets spanning four different scenarios. This alone is an achievement given the considerable number of strong, domain-specific competitors and the small margins that often separate published methods.  And critically, the second-best performing model shown in our experiments varies from case to case, implying that the gap between our single unified approach and the next best alternative framework fixed across all domains would be even larger (assuming such a model can somehow be successfully retrofitted across all four different scenarios).
>
> Also, there are quite a few cases where DIFFormer outperforms even domain-specific competitors by highly substantial margins, e.g., $69.24\pm 0.76$ (ours) against $65.57\pm 0.34$ (the best competitor) on Pokec in Table 3, $74.8\pm 0.5$ (ours) against $73.2\pm 0.4$ (the best competitor) on CIFAR in Table 4. And since our models are also efficient in practice (see Appendix E.5 for time/space comparisons), we believe the performance gains are arguably significant advances over prior art.
>
>
> **Q2: Clarification of the conditions for Theorem 1**
>
> To apply Theorem 1 to our particular DIFFormer instantiation in Sec. 4.1, the $\delta$ function is actually only required to be non-decreasing and concave on a restricted domain as opposed to all of $\mathbb R$; we have modified the description in Sec 4.1 to make this point more clear. Note that the layer normalization we use controls the norm of $\mathbf z_i$ to be within 1. Consequently, $z^2 = ||\mathbf z_i - \mathbf z_j||_2^2$ will lie between 0 and 2, and so $\delta(z^2)$ and $f(z^2)$ need only satisfy the Theorem 1 criteria within $[0, 2]$. Moreover, since the two specified forms of $f$ in Sec. 4.1 are positive and strictly decreasing w.r.t. $z^2$ in this restricted range, the corresponding $\delta$ will necessarily be concave and non-decreasing as required. Please see Appendix C and Fig. 3 where we provide the specific forms of $\delta$ for both choices of $f$; Fig. 3 also plots the corresponding curves for illustration purposes.
>
> In any event, per the above, Theorem 1 still holds without the assumption of quasiconvexity. Thanks to the reviewer for checking into the technical details of our analysis, which helps us to clarify the message.
>
>
> **Q3: Novelty**
>
> In terms of novelty, the reviewer suggested that related diffusion models have been studied in physics, which could somewhat impact the overall novelty. However, while we fully defer to the mathematical physics community for the origination of many diffusion-related ideas, there is nonetheless ample room for innovation when transferring to the ML context, particularly within the narrower class of neural architectures which is our focus. Indeed, there is no prior work we are aware of demonstrating how principled neural network layers can be constructed from a diffusion process that minimizes an underlying structure-regularized energy function. That being said, as we are admittedly not experienced in mathematical physics as the reviewer is, we would greatly appreciate any relevant background references that we could include to strengthen the foundation of our work.

---

### Official Review · Reviewer_tDZ1 · 2022-10-27

**Confidence:** 3
**Correctness:** 3
**Technical Novelty And Significance:** 3
**Empirical Novelty And Significance:** 3
**Recommendation:** 8

**Clarity, Quality, Novelty And Reproducibility:**

The paper was well written. Additional discussions could make the paper stronger.  E.g., could the authors also compare the running time for different methods on the tasks in sections 5.3 and 5.4?

**Strength And Weaknesses:**

Strength:

In this paper, the authors proposed a geometric approach to model sample points on the data manifold through diffusion on graphs. The proposed approach and methodology are interesting.  Moreover, the method achieved improved performance on multiple datasets.


Weakness:

The author could give more detail about the difference between the proposed method and label propagation used for semi-supervised learning.  How does the data sample graph affect the performance of the method?

**Summary Of The Paper:**

The authors introduce an energy-constrained diffusion model which encodes a batch of instances from a dataset into evolutionary states that progressively incorporate other instances’ information by their interactions. In their method, the diffusion process is constrained by descent criteria w.r.t. a principled energy function that characterizes the global consistency. Experiments show that the model could achieve superior performance in various tasks.

**Summary Of The Review:**

The method proposed in the paper is novel, and the writing and structure of the paper are good.

---

> ### Author Response · Authors · 2022-11-14
> **Response to Reviewer tDZ1**
>
> Thanks for the valuable comments and constructive advice. We are encouraged that the reviewer deemed our work novel, well written and significant.
>
> **Q1: Comparison between the proposed model and label propagation.**
>
> There is a distinct difference between our model and label propagation (LP), a classic semi-supervised learning (SSL) approach that is used as a baseline in our Table 2. LP considers propagating node *labels* along a *fixed input graph* topology to infer the labels for unlabaled data, while DIFFormer propagates node *embeddings* along an *adaptive layer-specific latent graph* topology that is optimized together with the model.
>
> There are also some SSL algorithms using pseudo labels as extra training signals and this line of work is at times referred to as label propagation in the literature as well. In contrast with them, DIFFormer is only trained on a supervised loss w.r.t. labeled data and does not use extra data augmentation with pseudo labels or a consistency-based regularization loss. These strageties are orthogonal to our contributions (a new encoder backbone). Even so, as demonstrated by new experimental results, incorporating such pseudo labels with DIFFormer can indeed provide further performance gains, e.g., improving the accuracy of DIFFormer-s on CIFAR-10 from $76.6 \pm 0.3$ to $77.1 \pm 0.3$ (see response to Q4 of Reviewer ckRq for more details).
>
> **Q2: How does the data sample graph affect the performance?**
>
> The importance of the input graph (when available) will generally vary across different tasks. For example, in our experiment on temporal dynamics prediction (Table 5), we found that the input graph lacks importance (likely because it is poorly aligned with the true latent structure of the data). Still, there are other cases where the input graph plays an important role, e.g., on node classification datasets (Table 2), and can help the model to better capture the underlying data geometry.
>
>
> **Q3: More comparison of the running time for Sec 5.3 and 5.4.**
>
> In Appendix E.5 we have presented some run-time comparisons on two node classification datasets Cora and Pokec used in Sec 5.1 and 5.2, respectively.  However, because our method is quite general and can apply equally well to considerably broader application scenarios, the reviewer makes a good suggestion that further timing evaluations are warranted.
>
> To this end, we added additional comparisons on the image dataset STL-10 and the spatial-temporal dataset WikiMath, as the representatives for datasets used in Sec 5.3 and 5.4, respectively, echoing the wide applicability of our method beyond node classification. The new results are updated in Table 9, with companion discussion included in Appendix H.5. The results consistently show that DIFFormer-s has good efficiency (even much faster than GAT on the large dataset) due to its linear complexity.

---

### Official Review · Reviewer_Hw3e · 2022-10-31

**Confidence:** 4
**Correctness:** 4
**Technical Novelty And Significance:** 4
**Empirical Novelty And Significance:** 4
**Recommendation:** 10

**Clarity, Quality, Novelty And Reproducibility:**

As mentioned above, I found the writing very clear. I think the work is of the highest quality, is sufficiently novel, and felt the model was explained in enough technical detail to be reproduced from the paper alone. (The PyTorch-style pseudo-code is a welcome addition in the appendix, but I assume the authors will also release their code for complete reproducibility.)

**Strength And Weaknesses:**

This paper is very strong in many respects. The theoretical framework fits well in backdrop of a larger ongoing effort to unify various machine learning models. The motivation and presentation is very clear throughout, both from a technical/mathematical sense as well as general writing style. The theorem which is proved is not just a nice-to-have, but rather fundamental to the solution, and the authors emphasize this in their presentation beautifully. I have no major weaknesses to report.

**Summary Of The Paper:**

This paper proposes a theoretical framework which leverages a diffusivity function $\mathbf S_{ij}$ to capture interdependencies in input data. In the extremes, one can consider an IID assumption - in which case the diffusivity function is just the constant identity matrix, and the model simplified to an MLP - or a setting where the inputs are vertices in a known graph - in which case using a diffusivity function $\mathbf S_{ij}$ with nonzero weights between edges $(i,j) \in \mathcal E$ can result in either a GCN or GAT, depending on the choice of weights.

Beyond incorporating existing architectures in a unified framework, this perspective also suggests that the diffusivity function can be computed based on the input features. The authors introduce the crucial idea of minimizing an energy function which measures the quality of instance states at a given step, and then prove the existence of a diffusivity function which guarantees that the energy of the instances will be minimized in each step. This result has one remaining design decision - the specification of a non-negative decreasing function - and the authors explore two obvious choices for this function, leading to a "simple diffusivity model" DIFFormer-s which is linear in terms of the number of inputs, and an "advanced diffusivity model" DIFFormer-a, the latter of which is capable of capturing more complex latent geometry but has quadratic complexity in the number of inputs.

A large array of experiments, both synthetic and on real data, are included, with an array of strong baselines. The authors also perform a comprehensive ablation study to assess various aspects of the design decisions.

**Summary Of The Review:**

This was an outstanding work, very polished and complete. Definitely ready for publication.

---

> ### Author Response · Authors · 2022-11-14
> **Response to Reviewer Hw3e**
>
> Thanks for the thorough review and comprehensive summary of our contributions on both theoretical and empirical fronts. In fact, the reviewer's summary quite accurately distills our intended message, and we are glad that the novelty and potential impact were well-received. And definitely, we will release the full version of our code upon publication to facilitate reproducibility.

---

### Decision · Program_Chairs · 2023-01-20

**Decision:**

Accept: notable-top-25%

**Justification For Why Not Higher Score:**

see above

**Justification For Why Not Lower Score:**

see above

**Metareview: Summary, Strengths And Weaknesses:**

Overall, all of the reviewers enjoyed the paper and found it had good merit and good novelty. As an AC, I also enjoyed the responses by the authors, clear and concise. It is recommended if possible to try to fit some of the material from the rebuttal into the main paper or an appendix. The paper, I believe, should be accepted.

**Note From Pc:**

if the above contains the word "oral" or "spotlight" please see: "oral" presentation means -> notable-top-5% and "spotlight" means -> notable-top-25%. As stated in our emails, we are disassociating presentation type from AC recommendations

**Summary Of Ac-Reviewer Meeting:**

see above